# Introducing risk inequality metrics in tuberculosis policy development

M. Gabriela M. Gomes [1,2], Juliane F. Oliveira [2], Adelmo Bertolde[3], Diepreye Ayabina[1], Tuan Anh Nguyen[4], Ethel L. Maciel[5], Raquel Duarte[6], Binh Hoa Nguyen[4], Priya B. Shete[7] & Christian Lienhardt[8,9]

Global stakeholders including the World Health Organization rely on predictive models for developing strategies and setting targets for tuberculosis care and control programs. Failure to account for variation in individual risk leads to substantial biases that impair data interpretation and policy decisions. Anticipated impediments to estimating heterogeneity for each parameter are discouraging despite considerable technical progress in recent years. Here we identify acquisition of infection as the single process where heterogeneity most fundamentally impacts model outputs, due to selection imposed by dynamic forces of infection. We introduce concrete metrics of risk inequality, demonstrate their utility in mathematical models, and pack the information into a risk inequality coefficient (RIC) which can be calculated and reported by national tuberculosis programs for use in policy development and modeling.

[1] Liverpool School of Tropical Medicine, Liverpool L3 5QA, United Kingdom. [2] CIBIO-InBIO, Centro de Investigação em Biodiversidade e Recursos Genéticos, Universidade do Porto, Vairão 4485-661, Portugal. [3] Departamento de Estatística, Universidade Federal do Espírito Santo, Vitória, Espírito Santo 29075-910, Brazil. [4] National Lung Hospital, Hanoi 10000, Vietnam. [5] Laboratório de Epidemiologia, Universidade Federal do Espírito Santo, Vitória, Espírito Santo 29047-105, Brazil. [6] Faculdade de Medicina, and EPIUnit, Instituto de Saúde Pública, Universidade do Porto, Porto 4050-091, Portugal. [7] Division of Pulmonary and Critical Care Medicine, University of California San Francisco, San Francisco, CA 94110, USA. [8] Global TB Programme, World Health Organization, 1211 Geneva 27, Geneva, Switzerland. [9] Unité Mixte Internationale TransVIHMI (UMI 233 IRD – U1175 INSERM – Université de Montpellier), Institut de Recherche pour le Développement (IRD), Montpellier 34394, France. Correspondence and requests for materials should be addressed to M.G.M. G. (email: gabriela.gomes@lstmed.ac.uk)

Tuberculosis (TB) is a leading cause of morbidity and mortality worldwide, accounting for over 10 million new cases annually[1]. Although allusions are often made to the disproportionate effect of TB on the poorest and socially marginalized groups[2,3], robust metrics to quantify risk inequality in TB are lacking. Data reported by the World Health Organization (WHO), which mathematical models often rely on for calibrations and projections, are typically in the form of country-level averages that do not describe heterogeneity within populations. In keeping with the spirit of the Sustainable Development Goals agenda[4], we postulate that mathematical models that account for heterogeneity and inequality may best reflect the potential impact of TB prevention and care strategies in achieving disease elimination. Further, we hypothesize that disease incidence patterns in a population reflect unobserved heterogeneity and may be used to inform model development and implementation.

Variation in individual characteristics has a generally recognized impact on the dynamics of populations, and pathogen transmission is no exception[5]. In infectious diseases, heterogeneities in transmission have been shown to have specific effects on the basic reproduction number, $R_0$, in ways which are unique to these systems[6–10]. In TB, as in other communicable diseases, this approach motivated the proliferation of efforts to collect data on contact patterns and superspreading events, to unravel processes that may affect transmission indices and models. The need to account for variation in disease risk, however, is not unfamiliar in epidemiology at large, where so-called frailty terms are more generally included in models to improve the accuracy of data analysis[11]. The premise is that variation in the risk of acquiring a disease (whether infectious or not) goes beyond what is captured by measured factors (typically age, malnutrition, comorbidities, habits, social contacts, etc), and a distribution of unobserved heterogeneity can be inferred from incidence trends in a holistic manner. Such distributions are needed for eliminating biases in interpretation and prediction[12,13], and can be utilized in conjunction with more common reductionist approaches, which are required when there is desire to target interventions at individuals with specific characteristics.

Individual risk of infection or disease relates to a probability of responding to a stimulus and, therefore, direct measurement would require the recording of responses to many exposures to obtain the frequency at which the outcome of interest occurs. In TB, this is unfeasible due to the relatively low frequency of disease episodes and the extremely variable time period between exposure and disease development, but may be approximated by subdividing the population in sufficiently large groups and recording occurrences in each of them. Then incidence rates can be calculated per group, and ranked. Supplementary Fig. 1 illustrates the population of a hypothetical country comprising low and high-risk individuals distributed geographically (but dividing by age or income level, for example, applied singly or in combination, could also serve our statistical purposes). Forasmuch as individuals are nonuniformly distributed, disease incidence will vary between groups and carry information about variation in individual risks.

Here we adopt concepts and tools developed in economics to measure inequality in wealth, such as the Lorenz curve[14] and the Gini coefficient[15], and modify them into suitable indicators of disease risk inequality. We then calculate a risk inequality coefficient for three countries—Vietnam, Brazil, and Portugal, representing high to low TB burdens—and derive country-specific risk distributions to inform transmission models. The resulting models are applied to investigate the conditions for reducing TB incidence by 90% between 2015 and 2035, one of the targets set by the WHO's End TB Strategy[16]. The results differ significantly from those obtained by a homogeneous approximation of the same models. We find that by considering heterogeneity, control efforts result in a lower impact on disease burden, except in special circumstances which we highlight. More generally, we elucidate how model predictability relies on certain forms of heterogeneity but not others, and propose a practical scheme for summarizing inequality in disease risk to be used in modeling and policy development for TB and other diseases.

## Results

**Risk inequality coefficient (RIC).** Figure 1 depicts Lorenz curves[14] for TB occurrences in the populations of Vietnam, Brazil and Portugal structured by municipalities (level 2 administrative divisions), enabling the calculation of a Gini coefficient[15] that we refer to as the risk inequality coefficient (RIC) (Methods). To inform mathematical models of TB transmission with two risk groups[12,17], we discretize risk such that 4% of the population experiences higher risk than the remaining 96%. This cut-off is consistent with previous studies[17,18], although it could have been set arbitrarily as the procedure does not depend on how we discretize what is conceivably a continuous risk distribution. The Lorenz curves corresponding to the discretization, which are depicted by the dashed lines in Fig. 1a, are then used as an approximation to the original solid curves with the same RIC.

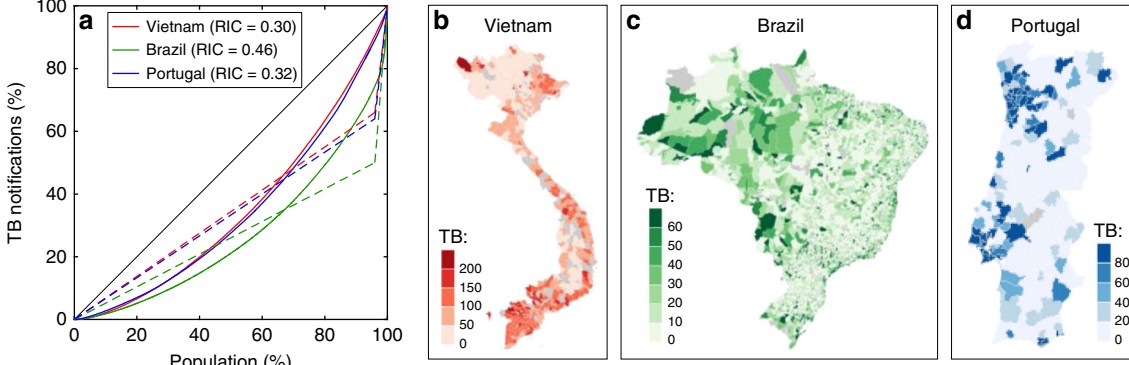

**Fig. 1** Risk inequality coefficient. **a** Lorenz curves calculated from notification data stratified by level 2 administrative divisions (697 districts in Vietnam; 5127 municipalities in Brazil; 308 municipalities in Portugal). A risk inequality coefficient (RIC) was calculated for each country from Lorenz curves as in Methods. Country maps with administrative divisions for Vietnam (**b**), Brazil (**c**), and Portugal (**d**), colored by number of cases notified per 100,000 person-years

**RIC-compliant transmission models**. Inequality in TB risk among individuals was implemented in three processes which were analyzed in alternation (Methods; parameters in Table 1): (i) contact rates; (ii) susceptibility to infection; and (iii) progression from primary infection to active disease. This study is primarily devoted to heterogeneity in contact rates, while the other two modalities are included for comparative purposes. Although the models differ in the precise implementation of the relative risk parameters ($\alpha_1$ and $\alpha_2$), in all three cases these can be calculated exactly and simultaneously with the mean effective contact rate ($\beta$), so as to match the country-specific incidence patterns reported for the first year in the data series.

The procedure was applied to data from Vietnam, Brazil, and Portugal (Fig. 2, for heterogeneous contact rates), resulting in risk variances of 10.5 in Vietnam, 11.1 in Brazil, and 5.63 in Portugal. Notice that these variances are consistently higher than the observed variances in TB incidence (2.3 in Vietnam, 5.1 in Brazil, and 2.7 in Portugal), indicating that transmission masks risk heterogeneity to some extent and we need to resort to models for

the inference of total variances[11]. Model outputs were then analyzed in-depth revealing a poor predictive capacity of homogeneous models and leading to the identification of acquisition of infection as the single most important process behind model disparities.

The risk distributions represented inside the various epidemiological compartments in Fig. 2b, e, h, are key to understanding why model outputs diverge. Mean risks have been normalized to one in all countries (i.e. the distributions in Fig. 2a, d, g have mean one), but as the system runs to endemic equilibrium high-risk individuals are infected predominantly. In other words, high-risk individuals are selected out of the uninfected compartment when a force of infection is in operation. As a result, the mean risk in the uninfected compartment decreases, decelerating the epidemic to the extent that the uninfected pool sustains transmission. This effect is greater for stronger forces of infection and larger risk variances, consistently with the mean risks displayed inside square brackets for the various epidemiological compartments. A similar process occurs

| Table 1 Parameters for tuberculosis transmission model | | |
|---|---|---|
| **Symbol** | **Definition** | **Value** |
| $\beta$ | Mean effective contact rate | estimated |
| $\mu$ | Death and birth rate | $1/80 \text{ yr}^{-1}$ |
| $\delta$ | Rate of progression from primary infection | $2 \text{ yr}^{-1}$ |
| $\phi$ | Proportion progressing from primary infection to active disease | 0.05 |
| $\omega$ | Rate of reactivation of latent infection | $0.0039 \text{ yr}^{-1}$ (Vietnam); $0.0013 \text{ yr}^{-1}$ (Brazil, Portugal) |
| $\tau$ | Rate of successful treatment | $2 \text{ yr}^{-1}$ |
| $\theta$ | Proportion clearing infection upon treatment | [0, 1] |
| $\alpha_i$ | Individual risk in relation to population average | estimated |
| $p_i$ | Proportion of individuals in low and high risk groups, respectively | $p_1 = 0.96$; $p_2 = 0.04$ |

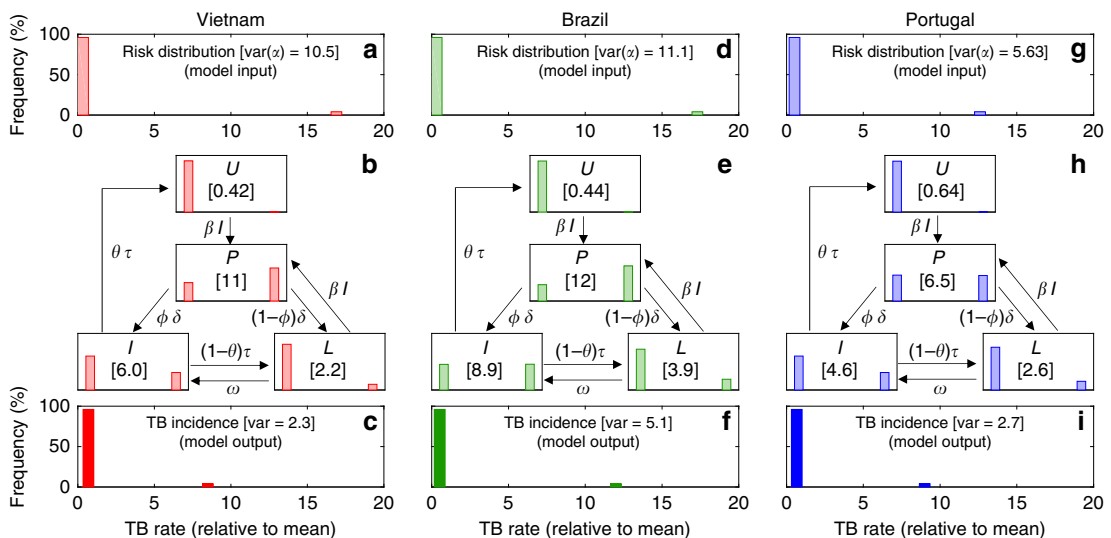

**Fig. 2** Tuberculosis transmission model with distributed contact rates. **a**, **d**, **g** Risk (contact rate) distributions inferred by fitting a mathematical model to notification data stratified in two risk groups (96% and 4% with risk factors $\alpha_1$ and $\alpha_2$, respectively) as in Methods ($\alpha_1 = 0.339$ and $\alpha_2 = 16.9$ in Vietnam [variance 10.5]; $\alpha_1 = 0.320$ and $\alpha_2 = 17.3$ in Brazil [variance 11.1]; $\alpha_1 = 0.516$ and $\alpha_2 = 12.6$ in Portugal [variance 5.63]). **b**, **e**, **h** Risk distributions in the various epidemiological compartments segregated by the transmission dynamics. Numbers in square brackets represent the mean baseline risk $\alpha$ among individuals populating each epidemiological compartment. **c**, **f**, **i** Distribution of incidence rates calculated from stratified model outputs ($Y_1 = 0.69$ and $Y_2 = 8.5$ in Vietnam [variance 2.3]; $Y_1 = 0.52$ and $Y_2 = 12$ in Brazil [variance 5.1]; $Y_1 = 0.67$ and $Y_2 = 9.0$ in Portugal [variance 2.7]). Model parameters as in Table 1. Clearance of infection upon successful treatment: $\theta = 1$. Country-specific parameter values: $\omega = 0.0039 \text{ yr}^{-1}$ and $\beta = 3.23 \text{ yr}^{-1}$ in Vietnam; $\omega = 0.0013 \text{yr}^{-1}$ and $\beta = 2.94 \text{ yr}^{-1}$ in Brazil; $\omega = 0.0013 \text{ yr}^{-1}$ and $\beta = 4.66 \text{ yr}^{-1}$ in Portugal. Notice that observed incidence variances $\langle (Y-1)^2 \rangle$ indicate underlying risk variances $\langle (\alpha-1)^2 \rangle$ which are consistently higher[11]

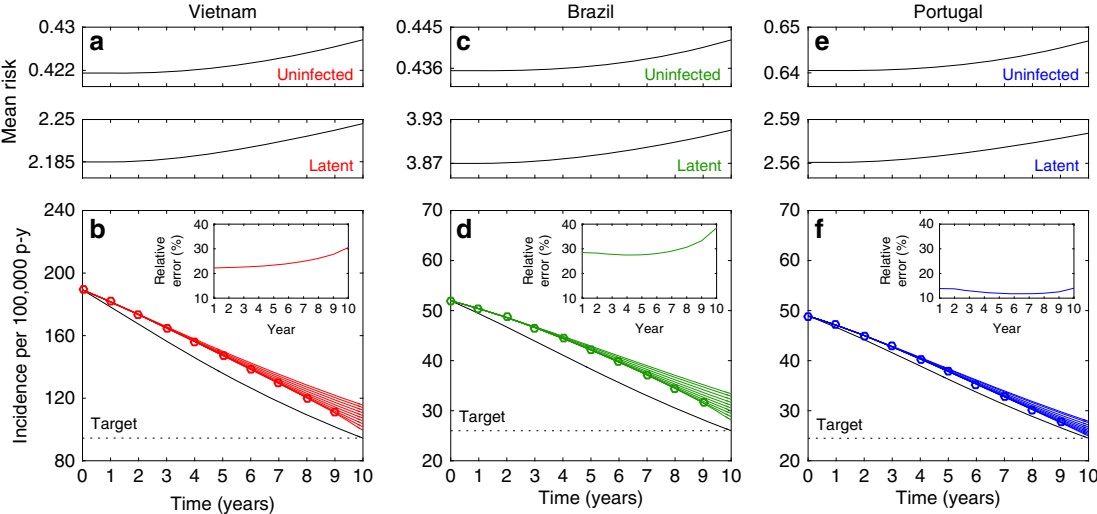

**Fig. 3** Moving targets. How (**b**, **d**, **f**) and why (**a**, **c**, **e**) fixed targets appear to be moving when observed from a homogeneous frame (Methods, and Supplementary Table 1). The model adopted in this illustration concerns heterogeneity in contact rates as governed by Eqs. (1)–(5). Mean risks among individuals in uninfected and latent compartments are calculated as $(U_1\alpha_1 + U_2\alpha_2)/(U_1 + U_2)$ and $(L_1\alpha_1 + L_2\alpha_2)/(L_1 + L_2)$, respectively. Model parameters as in Table 1. Clearance of infection upon successful treatment: $\theta = 1$. Country-specific parameter values: $\omega = 0.0039$ yr$^{-1}$, $\beta = 3.23$ yr$^{-1}$ (heterogeneous) or $\beta = 10.7$ yr$^{-1}$ (homogeneous) in Vietnam; $\omega = 0.0013$ yr$^{-1}$, $\beta = 2.94$ yr$^{-1}$ (heterogeneous) or $\beta = 17.3$ yr$^{-1}$ (homogeneous) in Brazil; $\omega = 0.0013$ yr$^{-1}$, $\beta = 4.66$ yr$^{-1}$ (heterogeneous) or $\beta = 17.1$ yr$^{-1}$ (homogeneous) in Portugal

for all epidemiological compartment where individuals are at risk of infection (i.e. uninfected ($U$) and latent ($L$) in the case of the model adopted here).

**Risk inequality as a compromiser of intervention impact**. The heterogeneous contact-rate model initiated according to 2002 incidences (Fig. 2) was run forward in time with a constant decay rate in reactivation to meet an arbitrary fixed target of halving the incidence in 10 years (Fig. 3b, d, f, black curves). If these estimations (exact calculations in this case) and projections had been made by the homogeneous model, the required control efforts would have been underestimated and the target systematically missed (Methods; Supplementary Table 1), with relative errors around 20–30% for Vietnam, 25–40% for Brazil, and 10–20% for Portugal (colored curves). This is because the force of infection decreases as the intervention progresses, reducing the strength of selection described above, which in turn allows for increasing mean risks in compartments at risk of infection (Fig. 3a, c, e), counteracting the intended effects of the intervention. Homogeneous models artificially disable this selection process, creating an illusion that control targets are moving when observed from a homogeneous frame.

This is a general phenomenon in infectious diseases, although there may be exceptional circumstances where the sign of the effect may be reversed as detailed below. In any case, it is a systematic error (bias) not to be confused with uncertainty in parameter estimates[19,20].

**Meeting WHO's End TB incidence targets**. The models were used to reproduce reported country-level trends for TB incidence in Vietnam, Brazil, and Portugal. Following initialization in 2002 as above, the model was fitted to the incidence declines reported by WHO until 2015. In the first instance we explored how much reactivation should have decreased had the observed incidence declines been attributed to changing this parameter alone at a constant rate (Supplementary Table 2). This was performed numerically by a binary search algorithm designed to meet 2015 incidences (Fig. 4). Trajectories were then prolonged until 2050 (dashed segments in the same figure) suggesting the need for

increased efforts to meet the End TB incidence targets (2035 targets marked by dotted lines). This initial exploration was completed by the introduction of a scale-up parameter ($\kappa$) to account for increased reductions in reactivation from 2020 onwards and estimating the necessary scaling to meet the 2035 target in each country (displayed as "$\times\kappa$" in the figure). As above, the homogeneous model consistently underestimates the required control efforts. In the following we refer to this as the *default* expectation when comparing the outcomes of the same investigation strategy applied to more realistic scenarios where incidence declines are attributed to a combination of parameters.

When incidence declines are attributed to reductions in the probability of progressing from primary infection to active disease ($\phi$, with the remaining $1 - \phi$ maintaining a latent infection) as well as reactivation ($\omega$), estimating the two decay rates is not possible with a simple binary search algorithm and we use a Bayesian Markov Chain Monte Carlo (MCMC) approach (Methods). Figure 5 depicts the declining annual incidences and model trajectories, based on the means and 95% credible intervals of the posterior distributions of decay rates in $\phi$ and $\omega$ (Supplementary Table 3), prolonged until 2020. Also in this scenario, control measures must be intensified for meeting the ambitious End TB targets. We apply the scaling factor $\kappa$ uniformly to the decay rates of the two parameters and estimate the required effort intensification. Heterogeneous contact-rate (Fig. 5a, c, e) and homogeneous (Fig. 5b, d, f) models are similarly effective at capturing the data, but require significantly different scale-up efforts (Supplementary Table 4). In contrast with the case where only reactivation was reduced, we now get an indication that Brazil requires less effort intensification under heterogeneity (in relation to that predicted by the homogeneous model) while Vietnam and Portugal comply with the default expectation. Inspection into the percent reduction curves for the two parameters reveals that scale-up tends to be more effective when the initial decline (pre-scale-up) is predominantly attributed to reducing reactivation (homogeneous in Vietnam and Portugal; heterogeneous in Brazil).

Under heterogeneous contact rates, the incidence declines observed in Vietnam and Portugal have been predominantly

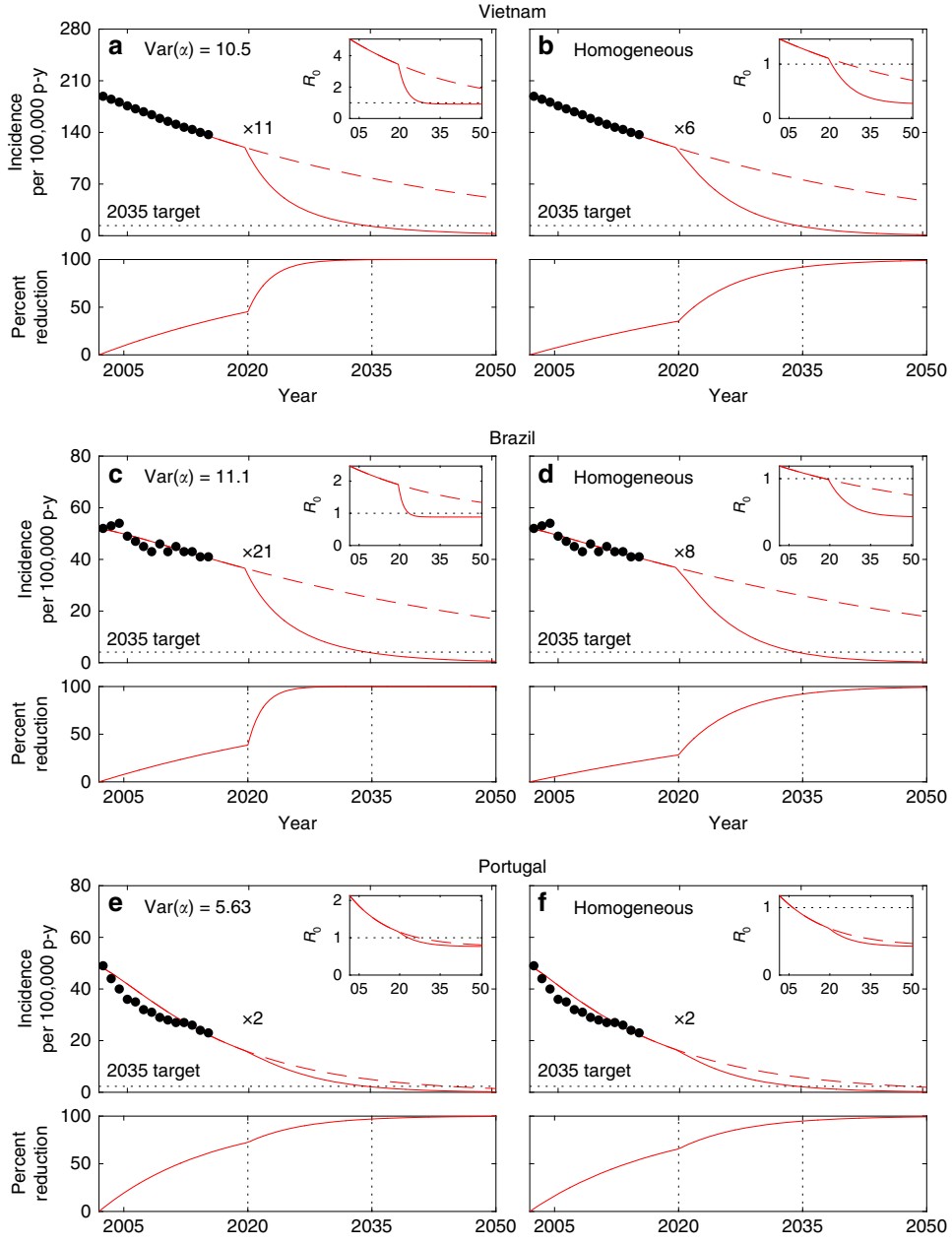

**Fig. 4** Model trajectories with heterogeneity in contact rates and gradual decline in reactivation ($\omega$). TB incidence from 2002 to 2015 (black dots) and model solutions under heterogeneous contact rates (**a**, **c**, **e**); homogeneous approximation (**b**, **d**, **f**). Initial parameters values calculated by adjusting the mean effective contact rates ($\beta$) to fit 2002 incidence rates: $\beta = 3.23 \text{ yr}^{-1}$ (**a**) or $\beta = 10.7 \text{ yr}^{-1}$ (**b**) in Vietnam; $\beta = 2.94 \text{ yr}^{-1}$ (**c**) or $\beta = 17.3 \text{ yr}^{-1}$ (**d**) in Brazil; $\beta = 4.66 \text{ yr}^{-1}$ (**e**) or $\beta = 17.1 \text{ yr}^{-1}$ (**f**) in Portugal. Incidence declines towards 2015 attributed to reducing reactivation: $\omega(t) = \omega_0 e^{r_\omega(t-2002)}$ (where $\omega_0 = 0.0039$ in Vietnam and $\omega_0 = 0.0013$ in Brazil and Portugal), with constant rates $r_\omega$ adjusted to meet the incidences observed in 2015 (Supplementary Table 2). From 2020 onwards, the trajectories split to represent two scenarios: rates of parameter change are maintained (dashed); scale $r_\omega$ by a factor $\kappa$ (represented as "×$\kappa$") to meet WHO incidence targets for 2035 (solid). The bottom plots in each panel represent the cumulative reductions in reactivation required to meet the targets calculated as $\hat{\omega}(t) = 1 - \omega(t)/\omega(2002)$. Clearance of infection upon successful treatment: $\theta = 1$. Other parameters as in Table 1. Model described by Eqs. (1)–(5), and $R_0$ given by (6)

attributed to reducing progression to disease from recent infection (Fig. 5a, e; bottom panels show blue curve above red in pre-scale-up phase). Given the assumption of identical scaling factors for both processes, the reduction in $\phi$ (blue) reaches saturation soon after scale-up is initiated leaving most of the remaining effort to $\omega$ (red) and inflating the required scaling efforts. Contrastingly, in Brazil the incidence decline has been largely attributed to reducing disease arising from reactivation (Fig. 5c; bottom panel shows red curve above blue pre-scale-up)

leaving the reduction in $\phi$ far from saturation and creates a scenario where reducing progression maintains substantial potential to generate further impact after scaling.

Naturally, there is no reason for scale-up factors to be the same for the two processes, and this result suggest that new ways to reduce reactivation are needed in Vietnam and Portugal. In relation to that, it also raises the importance of understanding what may have led to the declining reactivation rates in Brazil and how might other countries achieve similar goals. More detailed

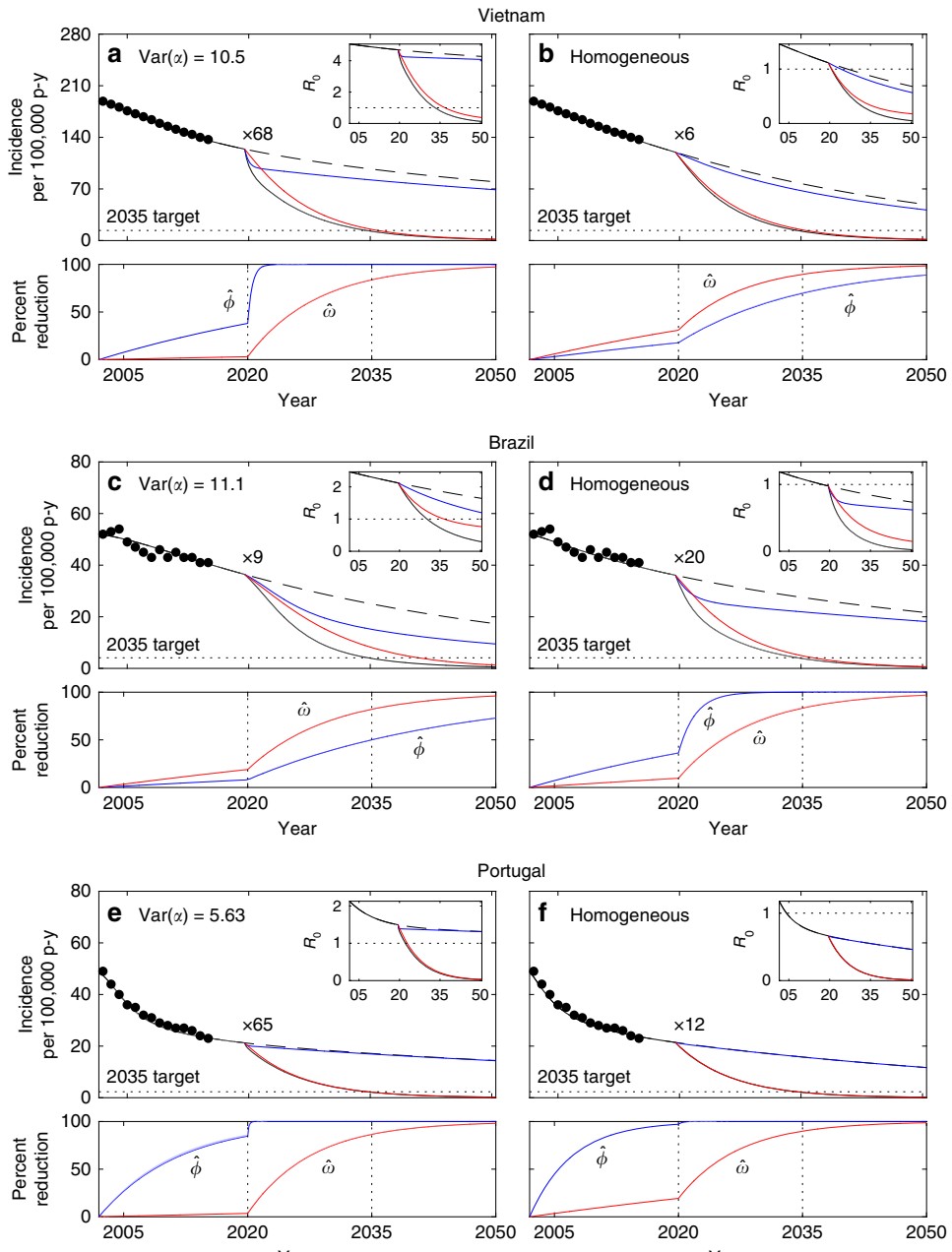

**Fig. 5** Model trajectories with heterogeneity in contact rates and gradual declines in disease progression ($\phi$) and reactivation ($\omega$). TB incidence from 2002 to 2015 (black dots) and model solutions under heterogeneous contact rates (**a**, **c**, **e**); homogeneous approximation (**b**, **d**, **f**). Initial parameters values calculated by adjusting the mean effective contact rates ($\beta$) to fit 2002 incidence rates: $\beta = 3.23$ yr$^{-1}$ (**a**) or $\beta = 10.7$ yr$^{-1}$ (**b**) in Vietnam; $\beta = 2.94$ yr$^{-1}$ (**c**) or $\beta = 17.3$ yr$^{-1}$ (**d**) in Brazil; $\beta = 4.66$ yr$^{-1}$ (**e**) or $\beta = 17.1$ yr$^{-1}$ (**f**) in Portugal. Incidence declines towards 2015 attributed to reducing disease progression and reactivation: $\phi(t) = 0.05e^{r_\phi(t-2002)}$ and $\omega(t) = \omega_0 e^{r_\omega(t-2002)}$ (where $\omega_0 = 0.0039$ in Vietnam and $\omega_0 = 0.0013$ in Brazil and Portugal), with constant rates $r_\phi$ and $r_\omega$ estimated using MCMC (Supplementary Table 3). From 2020 onwards, the trajectories split to represent four scenarios: rates of parameter change are maintained (dashed black); scale $r_\phi$ and $r_\omega$ by a factor $\kappa$ (represented as "$\times\kappa$") to meet WHO incidence targets for 2035 (solid black); apply the same scale up efforts to $r_\phi$ only (blue) or $r_\omega$ only (red). The bottom plots in each panel represent the cumulative reductions in disease progression and reactivation required to meet the targets calculated as $\hat{\phi}(t) = 1 - \phi(t)/\phi(2002)$ and $\hat{\omega}(t) = 1 - \omega(t)/\omega(2002)$, respectively. Clearance of infection upon successful treatment: $\theta = 1$. Other parameters as in Table 1. Model described by Eqs. (1)–(5), and $R_0$ given by (6)

datasets should be interrogated in search for answers, but this is potentially due to especially intense social protection programs implemented over recent decades in Brazil[21–25], leading to improved health conditions in population segments classically more at risk for TB.

The parameters that have been most commonly varied to explain incidence trends in modeling studies are rates of successful treatment ($\tau$) and mean effective contacts ($\beta$)[26]. For completion and comparability with other studies we conceive additional scenarios where the observed declines in incidence are attributed to decays in $\tau$ and $\omega$ (Supplementary Fig. 2 and Supplementary Tables 5 and 7) or $\beta$ and $\omega$ (Supplementary Fig. 3 and Supplementary Tables 6 and 7), and infer the respective attributions as above. In both cases the scaling in control efforts required to meet End TB incidence targets appears lower under heterogeneity. This seems counter-intuitive at first but see the

values of $R_0$ plotted as insets in Figs. 4, 5 and Supplementary Figs. 2, 3. During the scale-up phase, this transmission index is consistently below one in the homogeneous implementation and above one when heterogeneity is considered. Since $\tau$ and $\beta$ relate to ongoing transmission, scaling changes in these parameters is not effective at reducing incidence when $R_0 < 1$ and, consequently, the homogeneous implementations must rely on the reduction in $\omega$ alone to meet the targets. This process results in the inflation of the scale parameter $\kappa$ observed under homogeneity and reversion of the default expectation. The sensitivity of our conclusions to which parameters are actually varying in each setting reinforces the need for more discriminatory data and dedicated studies.

Results presented so far addressed heterogeneity in contacts rates, which implicitly considers that acquisition of infection is positively correlated with transmission to others[5,8–10,12,18]. But irrespective of how present heterogeneity in contact rates is in TB dynamics, there is a myriad of biological factors which contribute to making individuals different and may affect TB incidence patterns.

Figure 6 (and Supplementary Table 8) shows the results obtained by employing the same procedures as in Fig. 5 but assuming that heterogeneity affects susceptibility of infection given exposure, rather than the rate of contacts. The two variants

are in fact described by the same model, except for how the force of infection is formulated (Methods). Essentially, if we write the force of infection as $\lambda = \beta(\rho_1 I_1 + \rho_2 I_2)$, where the new parameters $\rho_1$ and $\rho_2$ represent the relative infectivities of individuals in risk groups 1 and 2, respectively, heterogeneity in contact rates[12] is retrieved when $\rho_i = \alpha_i$ and heterogeneity in susceptibility[17] is obtained by imposing $\rho_i = 1$, while a combination of the two would correspond to values in between.

The agreement between Figs. 5 and 6 supports the notion that the results are mostly insensitive to whether heterogeneity affects primarily contact rates or susceptibility to infection, but the case of Vietnam deserves a special note. Under the heterogeneous susceptibility formulation, the contribution of reducing reactivation to the decline in incidence is more evident than under heterogeneous contact rates (Fig. 6b). As a result the scaling factor required to meet the 2035 incidence target is substantially reduced. This is not sufficient to reverse the default conclusion that the homogeneous model underestimates control efforts (as it happens again in Brazil), but it brings the estimated scaling factor closer to that estimated by the homogeneous model. It follows that any combination of the two forms of heterogeneity is expected to lead to the same qualitative conclusions, whereas, quantitatively, the findings for Brazil and Portugal are confined to narrow ranges while for Vietnam they are highly sensitive to how

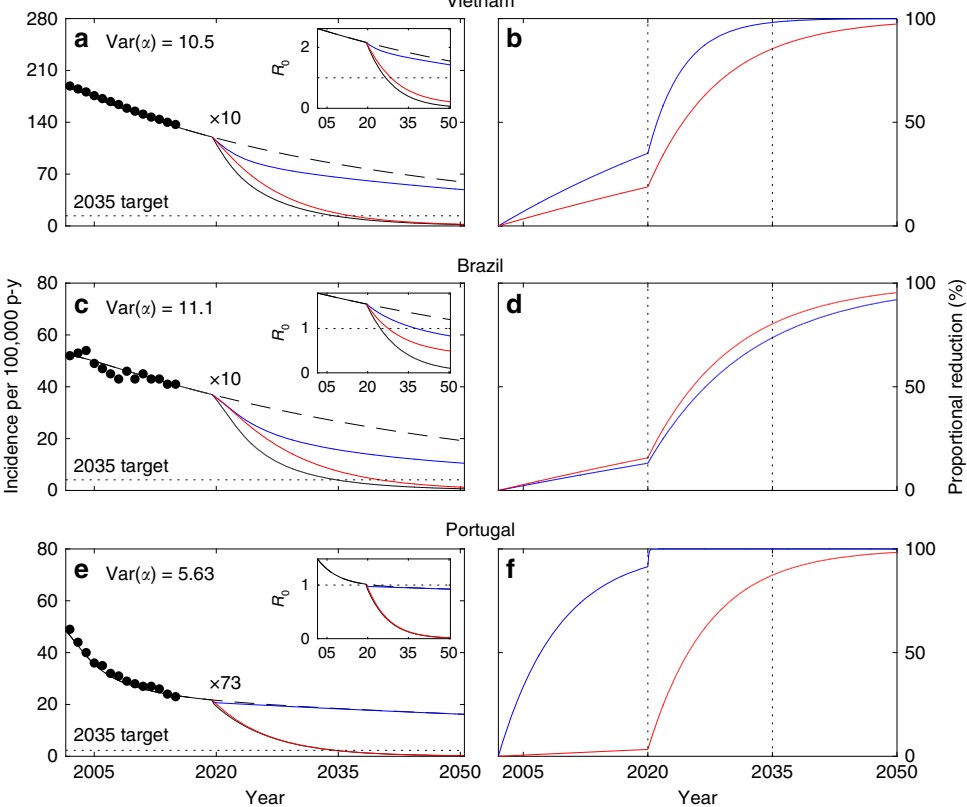

**Fig. 6** Model trajectories with heterogeneity in susceptibility to infection and gradual declines in disease progression ($\phi$) and reactivation ($\omega$). TB incidence from 2002 to 2015 (black dots) and model solutions under heterogeneous susceptibility to infection (**a**, **c**, **e**); cumulative reductions in disease progression and reactivation required to meet End TB incidence targets (**b**, **d**, **f**), calculated as $\hat{\phi}(t) = 1 - \phi(t)/\phi(2002)$ and $\hat{\omega}(t) = 1 - \omega(t)/\omega(2002)$, respectively. Initial parameter values calculated by adjusting the mean effective contact rates ($\beta$) to fit 2002 incidence rates: $\beta = 19.2\,\mathrm{yr}^{-1}$ in Vietnam (**a**); $\beta = 26.1\,\mathrm{yr}^{-1}$ in Brazil (**c**); $\beta = 21.6\,\mathrm{yr}^{-1}$ in Portugal (**e**). Incidence declines toward 2015 attributed to reducing disease progression ($\phi$) and reactivation ($\omega$): $\phi(t) = 0.05e^{r_\phi(t-2002)}$ and $\omega(t) = \omega_0 e^{r_\omega(t-2002)}$ (where $\omega_0 = 0.0039$ in Vietnam and $\omega_0 = 0.0013$ in Brazil and Portugal), with constant rates $r_\phi$ and $r_\omega$ estimated using MCMC (Supplementary Table 8). From 2020 onwards, the trajectories split to represent four scenarios: rates of parameter change are maintained (dashed black); scale $r_\phi$ and $r_\omega$ by a factor $\kappa$ (represented as "$\times\kappa$") to meet WHO incidence target for 2035 (solid black); apply the same scale up efforts to $r_\phi$ only (blue) or $r_\omega$ only (red). Clearance of infection upon successful treatment: $\theta = 1$. Other parameters as in Table 1. Model described by Eqs. (1)–(4) and (7), and $R_0$ given by (8)

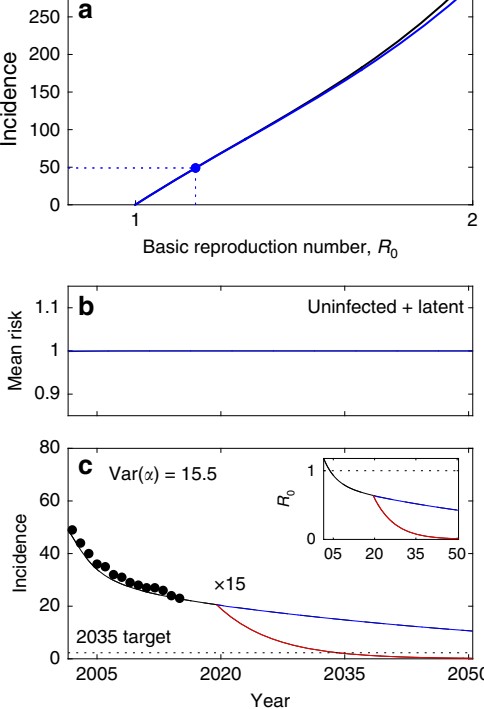

**Fig. 7** Model trajectories with heterogeneity in disease progression and gradual declines in progression ($\phi$) and reactivation ($\omega$). TB incidence from 2002 to 2015 in Portugal (black dots) and model solutions under heterogeneous progression to disease (**c**); mean risk (progression fraction) among susceptible individuals $[(U_1(t) + L_1(t))\alpha_1 + (U_2(t) + L_2(t))\alpha_2]/(U_1(t) + L_1(t) + U_2(t) + L_2(t))$ (**b**), and endemic equilibrium parameterized by the mean effective contact rate ($\beta$) plotted in terms of $R_0$ for the heterogeneous (blue) and homogeneous (black) models (**a**). Initial parameter values calculated by adjusting $\beta$ to fit 2002 incidence rates as shown in **a**: $\beta = 17.1\,\mathrm{yr}^{-1}$. Incidence declines toward 2015 attributed to reducing disease progression ($\phi$) and reactivation ($\omega$): $\phi(t) = 0.05 e^{r_\phi(t-2002)}$ and $\omega(t) = \omega_0 e^{r_\omega(t-2002)}$ (where $\omega_0 = 0.0039$ in Vietnam and $\omega_0 = 0.0013$ in Brazil and Portugal), with constant rates $r_\phi$ and $r_\omega$ estimated using MCMC (Supplementary Table 9). From 2020 onwards, the trajectories split to represent four scenarios: rates of parameter change are maintained (dashed black); scale $r_\phi$ and $r_\omega$ by a factor $\kappa$ (represented as "×$\kappa$") to meet WHO incidence target for 2035 (solid black); apply the same scale up efforts to $r_\phi$ only (blue) or $r_\omega$ only (red). Clearance of infection upon successful treatment: $\theta = 1$. Other parameters as in Table 1. Model described by Eqs. (9)-(13), and $R_0$ given by (14)

individual predisposition to acquire infection correlates with propensity to infect others. In any case, all the results presented so far imply heterogeneity in acquisition of infection.

The results presented are in stark contrast with forms of heterogeneity that do not affect acquisition of infection. Figure 7 (and Supplementary Table 9) shows that when heterogeneity is in the probability of progression from primary infection to active disease, model outputs do not deviate from the homogeneous implementation. This is because progression is not under the selection mechanisms described earlier in the paper, as demonstrated by the mean risk among susceptible compartments remaining flat at the value one (Fig. 7b) by contrast with what has been noted under heterogeneous contact rates, for example (Fig. 3a, c, e). Similarly, heterogeneity in rates of reactivation or treatment success should generally not lead to different model outputs unless correlated with predisposition for acquiring infection. This confirms our earlier premise that variation in

acquisition of infection is the single most important process behind the disparities between homogeneous and heterogeneous models, and hence the most important to estimate.

In further account to sensitivity analysis we show that the original results of Fig. 5 are robust to whether individuals clear the infection upon treatment or maintain a latent infection (Supplementary Fig. 4 and Supplementary Table 10).

**Prevalence of latent TB infection**. Prevalence of latent TB infection (LTBI) calculated from model trajectories generated by our heterogeneous models (27.0–28.9% in Vietnam, 15.2–16.1% in Brazil, and 16.9–18.0% in Portugal, in 2014; Supplementary Table 11) are generally consistent with estimates from a recent study[27]. This is irrespective of whether heterogeneity is in contact rates or susceptibility to infection. Even though these percentages are somewhat smaller than those expected under the homogeneous model, the reservoir must nevertheless be contained in all three countries if incidence targets are to be met.

## Discussion

The notion that heterogeneity affects the results of population models and analyses is not new[5,28–32], but we still face a general inability to measure it. We propose a concrete way forward for infectious disease transmission models, which is based on routinely collected data. Measures of statistical dispersion (such as Lorenz curves[14] and Gini coefficients[15]) are commonly used in economics to represent the distribution of wealth among individuals in a country and to compare inequality between countries, but rarely used in epidemiology[33,34]. Measuring disease risk of an individual is less direct than measuring income, but surely this can be overcome in creative ways for classes of diseases.

We have focused on tuberculosis, and shown how to approximate distributions of individual risk from suitably structured disease notification and population data (Fig. 1; Supplementary Fig. 1), and how to summarize the information into a simple risk inequality coefficient (RIC = 0.30 in Vietnam, RIC = 0.46 in Brazil, and RIC = 0.32 in Portugal), analogous to the Gini coefficients calculated by the World Bank to describe inequality in the distribution of wealth (0.38 in Vietnam, 0.51 in Brazil, and 0.36 in Portugal). Because they are based on the use of disease estimates at the level of administrative divisions within countries, there are limits to the accuracy of the RIC estimates, especially due to misreporting, which may be more severe in some countries than others. Other uses of the Gini coefficient, however, face the similar limitations while the methodology is still used to drive policy and program decisions and is improved upon as better data and formalisms become available. Importantly, the availability of comparable inequality metrics in economics and health can pave the way to pertinent studies between income inequality and health and provide a basis for equity considerations in policy development[35], a major component of the Sustainable Development Goals agenda[4]. In addition, we have demonstrated how to input this information into tractable mathematical models and why this is essential to accuracy and predictive capacity of these decision-making tools.

The approach followed here is in sharp contrast with those based on explicit metapopulation models[36–38]. We use incidence data of a country stratified into its administrative (geographical) divisions as a means to infer variation in disease risk among individuals, rather than as a direct measure of variation between the divisions themselves. To highlight this distinction we built a metapopulation model consisting of two subpopulations (patches), each with its intrinsic individual variation, and constrain the outputs to be consistent with patch incidences (Methods; Supplementary Fig. 5), according to data from our study

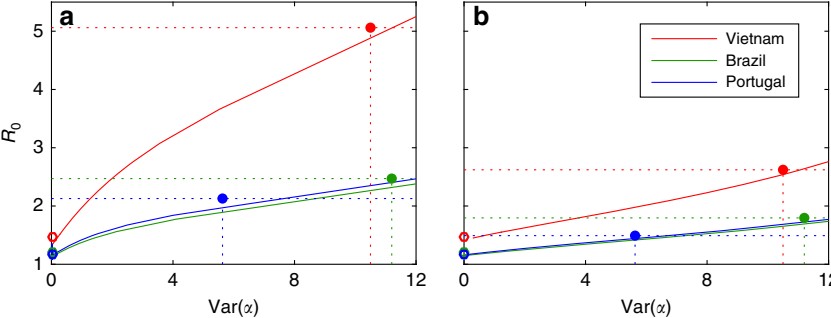

**Fig. 8** One-parameter family of metapopulation models. **a** Heterogeneous contact rates; **b** heterogeneous susceptibility to infection. Each point along a solid curve represents one model that produces country incidences in agreement with RIC values calculated in Fig. 1 (procedures described in Methods). Filled circles marks variances in individual risk and $R_0$ obtained for each country by the procedure utilized in this study, whereas open circles indicate $R_0$ estimated by homogeneous approximations

countries (Fig. 1). This sets a mathematical problem which can be solved over a range of country-level variances in individual risk (Supplementary Figs. 6 and 7), and for each variance there is an exact value of $R_0$ that makes the metapopulation model compatible with the stratified incidence data. The result is a curve describing $R_0$ as a function of variance in individual risk which is plotted in Fig. 8 together with the corresponding metrics obtained from the models used in this study (circles). The common practice of implementing a metapopulation without individual variation within subpopulations (lower limit of the curve), disables the action of selection at the individual level and carries similar biases to those present in homogeneous models (open circles). As individual variation increases, the curve approaches our heterogeneous models (filled circles), supporting the notion that the models proposed in this paper represent the dynamics of an average location within a country (with variation captured down to the individual level), in contrast with standard metapopulation models which describe an entire country structured into patches (with differentiation between patches but neglecting individual variation within).

Strikingly, the figure highlights an essential need for representing heterogeneity at the finest level if transmission indices are to be estimated accurately. In placing the models adopted here in the wider context of TB models with the same structure whose outputs are compatible with stratified incidence data for Vietnam, Brazil, and Portugal, the figure also reveals one potential limitation of the approach. The range of variances (and associated $R_0$ values) compatible with the data is wide and this is arguably the greatest current attrition to reaching high levels of certainty on parameters and predictions. This can be improved by combining multiple schemes for stratifying country incidence data alongside the development of more sophisticated methods for inferring variation in individual risk from patterns in the data.

In conclusion, the worldwide adoption of risk inequality metrics, such as the RIC proposed here or similar, has the potential to prompt an explosion of creativity in mathematical modeling, but it can also enable policymakers to assess risk inequality in each country, compare the metric across countries, and monitor the impact of equalization strategies and targeted interventions over time.

## Methods
**Lorenz curves and risk inequality coefficients.** Lorenz curves[14] are widely used in economics to calculate indices of inequality in the distribution of wealth, known as Gini coefficients[15]. Although rarely used in epidemiology, similar metrics can be adopted to describe inequalities in disease risk[33,34]. Here we construct a Lorenz curve for each study country from TB notifications and population data structured by municipalities (level 2 administrative divisions). Municipalities are ordered by incidence rates (from low to high) and cumulative TB notifications are plotted

against cumulative population (both in percentages). By construction, this results in a convex curve between (0, 0) and (100, 100), which would be a straight line in the absence of inequality. A risk inequality coefficient(RIC) can be calculated as the ratio of the area between the curve and the equality line, over the area of the triangle under the equality line. This gives a number between 0 and 1, which is analogous to the Gini coefficient commonly used to summarize income inequality, with the exception that while income can be measured at the individual level the assessment of TB risk cannot be made by analyzing individuals directly, but must be approximated from group measurements.

Supplementary Fig. 8 compares alternative Lorenz curves generated for Vietnam, Brazil and Portugal to explore the effects of timespan and group size. As we must comply with the administrative divisions already established in each country, level 2 appears to offer the best compromise between resolution (the smaller the units, the closer we get to measuring individual risk) and occurrences (the larger the units, the larger the numbers and the more accurate the risk discrimination[39]). Regarding timespan, the longer the data series the better. We used 10 years (2006–2015) in Vietnam and 14 years (2002–2015) in Brazil and Portugal to generate the respective RIC values.

We then use the RIC to inform risk distributions for TB transmission models. The Lorenz curves utilized to obtain RIC values consist of many segments (as many as administrative divisions; 696 in Vietnam, 5127 in Brazil, and 308 in Portugal). To keep our models tractable and low dimensional without compromising the overall variance in risk we construct two-segment Lorenz curves with the same RIC as the original and use this approximation to infer risk distributions for our TB models.

**Mathematical models.** We adopt a TB transmission model which is adapted from previously published studies[12,17], to represent risk heterogeneity in three alternative ways.

(i) Heterogeneity in contact rates:

$$\frac{dU_i}{dt} = q_i\mu + \theta\tau I_i - \lambda_i U_i - \mu U_i \tag{1}$$

$$\frac{dP_i}{dt} = \lambda_i(U_i + L_i) - (\delta + \mu)P_i \tag{2}$$

$$\frac{dI_i}{dt} = \phi\delta P_i + \omega L_i - (\tau + \mu)I_i \tag{3}$$

$$\frac{dL_i}{dt} = (1-\phi)\delta P_i + (1-\theta)\tau I_i - \lambda_i L_i - (\omega + \mu)L_i, \tag{4}$$

where subscripts $i = 1, 2$ denote low and high risk groups that individuals enter at birth in proportions $q_1$ and $q_2$, respectively. Within each group individuals are classified, according to their infection history, into uninfected ($U_i$), or infected in one of three possible states: primary infection ($P_i$); latent infection ($L_i$); and active tuberculosis disease ($I_i$) which is the infectious state. The model parameters along with their typical values used herein are listed in Table 1. The force of infection upon uninfected individuals is

$$\lambda_i = \frac{\alpha_i}{\langle\alpha\rangle}\beta(\alpha_1 I_1 + \alpha_2 I_2), \tag{5}$$

where $\alpha_i$ is a modifier of risk (contact rate in this case) of individuals in group $i$ in relation to the population mean $\langle\alpha\rangle = q_1\alpha_1 + q_2\alpha_2 = 1$, and the basic reproduction number is

$$R_0 = \frac{\langle\alpha^2\rangle}{\langle\alpha\rangle}\left[\frac{\omega + \mu}{\mu(\tau + \omega + \mu) + \theta\tau\omega}\right]\left[\frac{\phi\delta}{\delta + \mu} + \frac{(1-\phi)\delta\omega}{(\delta + \mu)(\omega + \mu)}\right]\beta, \tag{6}$$

where $\langle \alpha^2 \rangle$ is the second moment of the risk distribution, i.e. $\langle \alpha^2 \rangle = q_1 \alpha_1^2 + q_2 \alpha_2^2$. For simplicity we have assumed individuals to mix uniformly irrespectively of risk group.

(ii) Heterogeneity in susceptibility to infection:

When risk heterogeneity is attributed to susceptibility to infection the model is still written as in Eqs. (1)–(4), but the force of infection upon uninfected individuals becomes

$$\lambda_i = \alpha_i \beta (I_1 + I_2), \tag{7}$$

where $\alpha_i$ is the susceptibility of individuals in group $i$ in relation to the population mean $\langle \alpha \rangle = q_1 \alpha_1 + q_2 \alpha_2 = 1$. The basic reproduction number for this model is

$$R_0 = \langle \alpha \rangle \left[ \frac{\omega + \mu}{\mu (\tau + \omega + \mu) + \theta \tau \omega} \right] \left[ \frac{\phi \delta}{\delta + \mu} + \frac{(1 - \phi) \delta \omega}{(\delta + \mu)(\omega + \mu)} \right] \beta. \tag{8}$$

(iii) Heterogeneity in progression from primary infection to disease:

When risk heterogeneity is attributed to factors that affect the probability of progression from primary infection to active disease, the model takes the form

$$\frac{dU_i}{dt} = q_i \mu + \theta \tau I_i - \lambda U_i - \mu U_i \tag{9}$$

$$\frac{dP_i}{dt} = \lambda (U_i + L_i) - (\delta + \mu) P_i \tag{10}$$

$$\frac{dI_i}{dt} = \phi_i \delta P_i + \omega L_i - (\tau + \mu) I_i \tag{11}$$

$$\frac{dL_i}{dt} = (1 - \phi_i) \delta P_i + (1 - \theta) \tau I_i - \lambda L_i - (\omega + \mu) L_i, \tag{12}$$

with force of infection

$$\lambda = \beta (I_1 + I_2), \tag{13}$$

and $\phi_i = \alpha_i \phi$, representing the probability of progression from primary infection to disease for individuals in group $i$ in relation to the population mean $\langle \alpha \rangle = q_1 \alpha_1 + q_2 \alpha_2 = 1$. The basic reproduction number for this model is

$$R_0 = \left[ \frac{\omega + \mu}{\mu (\tau + \omega + \mu) + \theta \tau \omega} \right] \left[ \frac{\langle \alpha \rangle \phi \delta}{\delta + \mu} + \frac{(1 - \langle \alpha \rangle) \delta \omega}{(\delta + \mu)(\omega + \mu)} \right] \beta. \tag{14}$$

In all cases we use *risk* and *risk distribution* as generic terms to designate factors of variation in the predisposition of individuals to acquire infection or disease, which may be realized physically as rates of contacts with other individuals (i), or biologically as susceptibility to infection given exposure (ii) or progression to disease given infection (iii). We use the terminology *epidemiological compartment* to refer to the composite of all compartments for the same infection status (i.e. *uninfected* comprises both $U_1$ and $U_2$, etc.). We also introduce the notion of *mean risk* for each epidemiological compartment to track selection (e.g. the mean risk for $U(t)$ is calculated as $(U_1(t)\alpha_1 + U_2(t)\alpha_2)/(U_1(t) + U_2(t))$, etc.). We adopt two risk groups for concreteness, but formalisms with more groups would essentially support the same phenomena. Indeed, two recent studies implemented similar selection processes within populations structured into hundreds of risk groups[40,41].

The models accommodate an endemic equilibrium when $R_0 > 1$, as displayed by the solution curves parameterized by $\beta$ in Supplementary Figs. 9, 10 and Fig. 7a. Incidence rates in each risk group are approximated from model outputs by adding the positive terms in $dI_i/dt$ and dividing by the population in that group, i.e. $(\phi_{(i)} \delta P_i + \omega L_i)/q_i$ per year, and for the entire population as the weighted sum of these over risk groups.

**Model initialization**. Model trajectories are initialized assuming equilibrium conditions in 2002. Parameters describing the rates of birth and death of the population, the probability of progression from primary infection to active disease, and the rate of successful treatment, are set at the same values for the three countries: $\mu = 1/80 \; yr^{-1}$; $\phi = 0.05$ (ref. [42]), $\tau = 2 \; yr^{-1}$ (ref. [43]). The rate of reactivation is considered three times higher in South East Asian than in Western populations: $\omega = 0.0013 \; yr^{-1}$ in Brazil and Portugal; $\omega = 0.0039 \; yr^{-1}$ in Vietnam (ref. [44]). The mean effective contact rate ($\beta$) was calibrated to enable model solutions to meet country-level incidences estimated by the WHO for 2002 (Supplementary Figs. 9, 10 and Fig. 7a). Risk group frequencies are set at $q_1 = 0.96$ and $q_2 = 0.04$ and the relative risk parameters ($\alpha_1$ and $\alpha_2$) estimated as described below. The results are then displayed in terms of the non-dimensional parameter $R_0$, which is linearly related to $\beta$ according to Eqs. (6), (8), (14).

The same procedure was carried out for the mean field approximations of the respective models. At this point it can be confirmed that $R_0$ estimates are typically higher under heterogeneity[12]. We adopt heterogeneity in contact rates (i) as the default model throughout the paper, and use the susceptibility (ii) and disease progression (iii) variants for completion. Hence, unless specified otherwise, the results shown in the paper refer to heterogeneity in contact rates.

**Risk distributions**. Given a Lorenz curve (Fig. 1a), any discretization can be assumed to define how concentration of risk will enter the model. We adopt a division into 96% low-risk and 4% high-risk groups, but the procedure is not

specific to the chosen discretization. A distribution of incidences is then constructed as to produce the same RIC as the original curve: a segment $q_1 = 0.96$ of the population accounts for $(100 - y)$% of the incidence, while the remaining segment $q_2 = 0.04$ accounts for the remaining $y$% (Fig. 1a). The transmission model is solved as above, and the relative risk parameters $\alpha_i$ are calculated (Fig. 2a, d, g) so as to output the country-specific incidence distributions (see Fig. 2c, f, i). This was performed numerically by binary search to adjust the variance in the parameters $\alpha_i$ such that the variance in the output incidences agrees with the notification data.

Under any positive force of infection, the two risk groups segregate differently to populate the various epidemiological compartments, as depicted in Fig. 2b, e, h, resulting in mean risks that differ from one for specific compartments, and thereby deviating from homogeneous approximations. Crucially, the mean risks among individuals that occupy the various epidemiological compartments (square brackets in the figure) respond to dynamic forces of infection causing divergence from predictions made by homogenous models.

**Moving targets**. The model, with the estimated risk distributions, parameters, and initial conditions, fitting the 2002 incidences (189 in Vietnam, 52 in Brazil, and 49 in Portugal, all per 100,000 person-years), is run forward in time with a constant decline in reactivation rate as to meet an arbitrarily fixed target of halving the incidence in 10 years. As in the calculation of risk variance above, also here we refer to a simple numerical calculation performed by binary search. We write the reactivation rate as $\omega(t) = \omega(2002) e^{r_\omega (t - 2002)}$ per year, and approximate $r_\omega$ in order to meet the desired incidence target by year 2012.

Starting with initial reactivation rates of 0.0039 per year in Vietnam, and 0.0013 per year in Brazil, and Portugal, we find that meeting the target by this strategy alone, would require values of $r_\omega$ as specified in the heterogeneous column of Supplementary Table 1, or equivalently a decline in reactivation by $1 - e^{r_\omega}$ each year. This is to say that, in 10 years, the reactivation rates would have been reduced to values also shown in the respective column of Supplementary Table 1.

Suppose that these estimations and projections were being made by the mean field approximation of the same model, and the outcomes were monitored yearly and readjusted if necessary. The expectations would have been that lower absolute values would be required for the decay rate parameters $r_\omega$. Since the real population is heterogeneous, however, we simulate this decline for the first year with the heterogeneous model. The result is that, instead of achieving the incidences projected by the homogeneous model ("target" homogeneous column in Supplementary Table 1), the reality would lag behind ("achieved" homogeneous column in Supplementary Table 1), a result that the homogeneous model would attribute to insufficient effort exerted in reducing reactivation. From the homogeneous frame, an observer would have likely concluded that the decline had been lower due to some implementational failure, would have re-estimated the effort to meet the target over the remaining 9 years, now with an intensification to compensate for the lag of the first year. This process is simulated recursively for 10 years to populate Supplementary Table 1 and to generate Fig. 3. The insets in Fig. 3b, d, f, depict the relative error committed each year.

The dynamics of the mean risk of infection in the uninfected and latent compartments as the described interventions proceeds are shown in Fig. 3a, c, e, to demonstrate the action of selection. This is the key process leading to the deviation between the homogeneous and heterogeneous models.

**Meeting End TB targets**. The model with initial conditions, parameters and distributions estimated for 2002, is used to reproduce reported country-level trends for TB incidence in Vietnam, Brazil, and Portugal. Incidence declines between 2002 and 2015, reported by WHO for each of the three countries, are assigned to changes in pre-specified parameters (here set as $\phi$ and $\omega$ for illustrative purposes but alternative combinations have also been used). The decline is shared among the selected parameters as estimated below.

As incidence declines we monitor the reductions being made on each parameter, namely, on the probability of progression from primary infection to active disease $[1 - \phi(t)/\phi(2002)]$ and on the reactivation rate $[1 - \omega(t)/\omega(2002)]$.

**Parameter estimation**. Assuming that the incidence declines reported by WHO between 2002 and 2015 for Vietnam, Brazil and Portugal are due to reducing $\phi$ and $\omega$ at constant rates ($r_\phi$ and $r_\omega$, respectively), resulting in exponentially decaying parameters such that $\phi(t) = \phi(2002) e^{r_\phi (t - 2002)}$ and $\omega(t) = \omega(2002) e^{r_\omega (t - 2002)}$, we proceed to estimate $r_\phi$ and $r_\omega$. We used a Bayesian Markov Chain Monte Carlo (MCMC) approach to find posterior sets of these decay rates. We assume Gaussian priors and base our likelihood on the weighted squared error function

$$\chi^2 = \sum_{i=1}^{n} \left( \frac{B_i^d - B_i}{\sigma_i^d} \right)^2 \tag{15}$$

where $B_i^d$ are the data points, $B_i$ are the model outputs, and $\sigma_i^d$ are the corresponding measurement errors. This is equivalent to using the likelihood ($L$) such that $\chi^2 = -2 \log(L)$, under the assumption of Gaussian noise[45,46]. In the absence of the sampling distribution for the data, the error variance is sampled as a conjugate prior specified by the parameters $\sigma_0$ and $n_0$ of the inverse gamma

distribution where $\sigma_0$ is the initial error variance and $n_0$ is assumed to be 1 (as larger values limit the samples closer to $\sigma_0$)[47]. We use the MATLAB MCMC package developed by Haario et al. (2006)[48]. We initially minimize the error function and use these local minima as initial values for the parameters in the MCMC run. We infer a MCMC chain of length $10^5$ and adopt a burn in of $2 \times 10^4$ after assessing the Gelman-Rubins-Brooks potential scale reduction factor (psrf) plots of the posterior distributions (see Supplementary Figs. 11, 12).

**Comparison with metapopulation models.** As implied by Supplementary Fig. 1, geographical units are not conceptualized as homogeneous patches but rather as harboring heterogeneity down to the individual level. The transmission dynamics represented in our models is that of a country's average patch (with variation in risk among individuals) rather than a metapopulation consisting of multiple patches (each occupied by a homogeneous population and variation in risk among patches). To highlight this essential distinction, we have constructed a metapopulation model consisting of two subpopulations (A and B), each characterized by a distribution in individual risk (Supplementary Fig. 5).

Subpopulations (or patches) in this toy model are composed of individuals drawn from a common pool of high and low-risk individuals (in proportions 4 and 96%, respectively), and what characterizes each patch is the fraction of its individuals who are high-risk (rather than introducing patch-specific effective contact rates, $\beta_A$ and $\beta_B$, explicitly as commonly practiced). We assume a single $\beta$ for the entire metapopulation and vary the proportion of individuals in A who are high risk ($q_{2A}$) and calculate the corresponding proportion in B ($q_{2B}$). Basically, we have a family of metapopulation models, parameterized by the proportion of high-risk individuals in one of the patches, that we can completely resolve to match the incidence and RIC for each of our study countries.

We calculate relevant measures, such as variance in individual risk at the level of the entire metapopulation and $R_0$. These two metrics are shown as functions of $q_{2A}$ in Supplementary Figs. 6 and 7 (for heterogeneous contact rates and heterogeneous susceptibility, respectively) and one versus the other in Fig. 8. Open and filled circles are added to Fig. 8 for comparison of the same metrics under the homogeneous and heterogeneous models used in this study.

For simplicity we did not include transmission between subpopulations in this exercise, but there is no reason to expect sudden changes in outcome when this is added.

**Reporting summary.** Further information on research design is available in the Nature Research Reporting Summary linked to this article.

## Date availability

Estimated country-level incidence obtained from the WHO's global tuberculosis database (http://www.who.int/tb/country/data/download/en/). Municipality-level notification and population data used in Fig. 1 provided by National Tuberculosis Programs.

## Code availability

Computer programs were written in MATLAB R2015b as detailed in Methods. Maps were produced with Map in Seconds (http://www.mapinseconds.com).

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

## Acknowledgements

The Bill and Melinda Gates Foundation is acknowledged for its support through grant project number OPP1131404. M.G.M.G. and J.F.O. received additional support from Fundação para a Ciência e a Tecnologia (IF/01346/2014), and M.G.M.G. and D.A. from the European Union's Horizon 2020 research and innovation programme under grant No 733174 (IMPACT TB).

## Author contributions

M.G.M.G., P.B.S., and C.L. designed the study; E.L.M., R.D., and B.H.N. provided data and expertise; M.G.M.G., J.F.O., A.B., D.A., and T.A.N. performed the analysis; M.G.M.G. drafted the manuscript; all authors revised and approved the final version.

## Additional information

**Competing interests:** The authors declare no competing interests.

