## [Peer Review File · Nature Communications]

Reviewers' comments:

Reviewer #1 (Remarks to the Author):

The authors describe heterogeneity in TB risk and demonstrate that failure to introduce this heterogeneity in the analysis may lead to models missing their predictions in terms of relative impact of interventions. They introduce a metric to approximate the distribution of risk within a population (ie risk inequality coefficient), to allow modellers to use available (routinely collected) data to inform the needed distributions in risk and as an indicator for policy makers to assess progress in equalisation strategies. They compare their analysis in three countries with different patterns in TB epidemiology.

Broadly, I find the article interesting for a general audience, especially those specialised in infectious disease control; however, in my opinion the authors tried to convey too many messages in the paper (methods, data availability, public health application, comparison across three countries) and, in doing so, it made their message muddled. I think the manuscript would benefit from refining the scope and focusing the message – is this a methodological or a public health message?

I find that the methods are appropriate and the authors have provided sufficient detail for other researchers to reproduce their work - if access is given to same NTP data. My only comment on the methods is that the authors compare future predictions using models with and without heterogeneity and find that these predictions differ. I am missing the uncertainty - are these predictions really different or is a homogeneous model a good-enough simplification of a heterogeneous one? A better way to test their hypothesis as to which model predicts better reality would be to compare with past trends.

In terms of the conceptual development in methods - the authors made a compelling point to advance way models represent populations when looking at TB transmission. As they point out, the concept of heterogeneity in risk affecting transmission and therefore the impact of interventions is not new. The novelty comes from trying to characterise this heterogeneity using routine data into a single metric. Would be interesting if the authors could place their approach in context - how does this method compare to efforts in other infectious diseases to characterise heterogeneity across geographies in transmission disease models, for example malaria and HIV? What are the advantages and intrinsic assumptions of using a GINI-like coefficient based on geographical distribution of risk?

The application of heterogeneity in risk concepts in models should be included for two main reasons: 1) failure to do so will lead to the wrong predictions of impact, and 2) it allows the monitoring of an indicator of inequality in risk distribution to track progress in equalising risk. The main assumption is that the aim of public health interventions is to equalise or at least change these distributions. However, not all risk inequalities are similar. Risk inequalities can arise for many reasons and can be considered fair (such as due to biological determinants) or unfair (such as due to poverty – this is mentioned in the introduction but it feels like a late addition and not an important point) from a public health perspective. Policy makers will want to monitor progress in equalising unfair distributions but may not want to allocate limited resources to other strategies focusing on equalising fair heterogeneity. In the paper, authors map risk distribution across districts. The implication is that geographical distribution of risk is unfair. But without knowing what is driving this heterogeneity in geographical distribution, its correlation with socio economic status or poverty for example. it is difficult to assess the public health value and application.

In terms of the public health message – I am not clear what was the advantage of applying this method to Vietnam, Brazil and Portugal. Was there an added value of cross country comparison? If so, it will be good to discuss that. I think the example of achieving TB targets in 2035 makes this work useful to policy makers; however, the assumptions on how to achieve the reductions needed in reactivation rates (among others) are unclear. The authors argue that it does not make a

difference how you model these reductions – the point is that the predictions will change if heterogeneity is added to the model. But in doing so, they also reduce the public health value of the analysis.

Overall – I think it is a progressive manuscript that will advance the knowledge in the field, but the message needs more clarity and purpose and the methods need to include uncertainty estimates.

Reviewer #3 (Remarks to the Author):

This work presents some interesting insights into the control of tuberculosis, and in the limitations of homogenous models in informing control priorities. However, I felt it was let down by being so hard to follow: I strongly suggest that the authors work to improve the clarity, in order to open this work to a public health audience, rather than limiting it to a specialist mathematical one.

If I understand correctly, the main messages of the paper are:

1. Using homogeneous transmission models for heterogenous populations can distort the control priorities indicated by these models.
2. The extent of these distortions (and thus the utility of homogenous models) depends on the types of heterogeneity that are in effect.
3. In cases where there is large heterogeneity in acquisition of infection, homogenous models can overstate the importance of certain interventions, while understating the importance of others. However, other types of heterogeneity (e.g. in infectiousness) may be more benign.
4. The authors present a method (analogous to Gini's coefficient in economics) for quantifying the amount of heterogeneity in a population structured in some way (e.g. geographically, or by some risk factor).

Main comments

The way the paper is currently structured, it appears that homogeneous models are inappropriate for any type of heterogeneity. However, it is only towards the end (p.8) that it becomes clear that this only applies under a very specific type of heterogeneity (in acquisition of infection). Under other types of heterogeneity, presented only in the supporting information, there is less distortion between homogenous and heterogeneous models. The authors should make this much clearer - perhaps bringing some of those supporting information figures to the main text, to enable a fair comparison of different types of heterogeneity. Wording also needs to be adjusted in places: for example in p.2 line 37, where 'unsuitable' should be 'potentially unsuitable'. Acknowledging this also highlights a key data need, understanding what type of heterogeneity is operating in a given population.

I found the references to 'cohort selection' much more confusing than clarifying: it needs to be either rewritten, moved to the supporting information, or removed altogether. My issue is that for TB, reinfection can and does occur. That is, why should it matter whether or not the disease preferentially attacks the 'high risk' group? They can be reinfected anyway. My suggestion is that this might be better explained in terms of the effective reproduction number: adverse changes in population susceptibility should, in principle, manifest through this metric. At the moment though, to me figure 3 does very little to clarify the dynamics.

The use of the Lorentz curve is an innovative and revealing approach to inequalities in TB. However, the authors should also acknowledge its limitations more fully: especially, that it is best suited for situations where almost all cases are notified - otherwise it would tend to obscure, not reveal, some important inequalities (principally in access to high-quality, TB-notifying services).

Other comments

Although the Gini coefficient is a fresh and interesting proposal, straightforward statistical tests also detect population variation in risk (e.g. routine generalized linear models). It will be helpful to clarify what, if at all, the Lorentz curve approach offers in addition to these straightforward methods. Otherwise, it seems to me to be a useful and helpful way of communicating (but not necessarily detecting) inequality in a population. There is also a need to state more clearly the limitations of the 'Lorentz approach', as described below.

Authors have calibrated both models (heterogeneous and homogeneous) to give the same incidence. Do the models also agree in prevalence? This would seem important as the prevalence/incidence combination determines the average duration of disease, which in turn can be critical for intervention priorities. Have the authors controlled for this variation by ensuring both models are matched by prevalence too?

In practice, contact heterogeneity can play an important role in interactions between 'high-risk' and 'low-risk' individuals - potentially weakening the effects demonstrated in this paper, in cases where individuals preferentially mix with others in their own risk group. It would be helpful to describe this as a limitation.

The manuscript would benefit greatly from editing to improve clarity and to help the reader. Some examples:

- Comments are made without explanation, e.g. reference to 'variance' without defining what variance is being calculated, and the 'conservative' nature of these estimates, without explaining why or in what sense they're conservative

- p.4, line 81: Suggest avoid mentioning the model here - this paragraph refers to the inequality metric that the authors propose, which later serves as an input for the modelling. Mentioning the model at this stage seems to confuse things.

- p.4, line 89: "...as the single most important process..." Suggest to reword this so it's clear that these are modelled scenarios we're talking about, rather than measured phenomena.

- p.5, line 109: Here and in the Methods, it is unclear what it means to 'share equally' incidence declines amongst different parameters.

- p.6, line 123: "...adjust the data undistinguishably well..." Suspect there's a typo here - did the authors intend to say 'capture the data equally well' or something to that effect?

- p.11, equation 5: until this point I had understood that the model was about heterogeneity in susceptibility to infection, but it now appears for the first time that it additionally includes heterogeneity in transmission. Again, this confuses things: to make the story more systematic (and easier to follow), the authors could, for example, present a range of different types of heterogeneity in the main text, showing a comparison between them. Or, in the main text, they might show the model with solely heterogeneity in susceptibility, and demonstrate the other mechanisms in the supporting info. The current approach, which mixes these approaches, is confusing.

- p.12, line 259: I'm not sure why annual incidence is being calculated from the point values of

P_i , in a non-equilibrium model - isn't there an integral term needed here?

- p.13, line 292: Crucial, but obscure - please clarify, or remove.

Figures and tables

Again, I found these outputs unnecessarily hard to follow.

Figure 1c. What do the histograms show? The caption suggests that the figure illustrates the distribution in parameters, but the histogram appears to be associated with the state variables. It is also unclear what is meant by 'segregated by the transmission dynamics calibrated by the 2002 incidence'.

Figure 2. Please add a legend to highlight at a glance which parameters the different colours represent, even if only in panel c, or any other panel with available whitespace. Moreover, why does the blue curve show such abrupt changes in incidence (decreasing and then levelling off) compared to the other curves? Also, please mention briefly in the figure caption, how the 'scale changes' were selected for each of the parameters after 2020.

Figure 3. Is this the same as the ARTI, or equivalently the force-of-infection? If so, why should latent and uninfected compartments suffer different forces-of-infection? I suppose this may have something to do with the fact that there tends to be an over-representation of high-risk individuals in the latent compartment - in which case it would be really helpful for the reader to mention this in the caption. Also, to aid comparison, it would help to label each country with a different colour, perhaps distinguishing the latent and uninfected by solid and dashed lines. I'm not sure what the current dashed lines add to the figure.

Extended Data Fig.2: What does this add to Figs 1 and 4? Could the ratio of A and B terms not be explained easily in Fig.1 instead?

Extended Data Fig.3: Please clarify by adjusting the caption to 'Risk inequality metrics under alternative administrative groupings' or similar.

Extended Data Fig.5: seems important to the story, but rather mysterious in its current form. What are the parallel diagonal lines in panel A? What are the circles? These don't seem to be described anywhere that is straightforward to find.

Extended Data Fig.6: What is a 'cumulative reduction' here - may be better to call this the 'reduction relative to 2020 value' or something similar?

- p.31, Extended data table 1: Please clarify what the semi-colons mean

- p.32, Extended data table 1: the reader is left to track down what r_w means - it would be helpful to add a brief description here.

Reviewer #4 (Remarks to the Author):

A very interesting paper, with simple yet powerful ideas written up in a clear way. The idea of a Gini coefficient, of the kind used routinely by economists, for infectious disease epidemiology is relevant and could potentially provide a very compact summary of the degree of heterogeneity in transmission seen in populations. The authors provide model based evidence to demonstrate how this measure could lead to significantly different calculations of quantities that are of enormous importance such as the estimated time required for success of an intervention program.

I do think that it's currently harder than it needs to be in the paper to see how to apply the Gini coefficient, and implied Lorenz curve, to heterogeneous risk group mathematical models in order to allow projections of the impact of interventions to be made. Figure 1 is good and almost contains enough information for a modeler to reconstruct the calculations of the authors, but it would be good if the authors could include an example calculation (perhaps in supplementary material) in which the parameters α_1 and α_2 are discovered via the process described in the methods section.

Another issue that deserves comment by the authors (if not a calculation) is whether a two subgroup model is a sufficiently finely-structured model to be used as a simple proxy for a heterogeneous model in which the full heterogeneity implied by the Lorenz curve is modeled (perhaps by a large number of population subgroups that would more accurately reflect the true susceptibility distribution). I'm not suggesting that such a model needs to be constructed by the authors here, but that there could be some explanation of the sufficiency of the two group model.

The idea of estimating the necessary intervention strength using an incorrect model and then allowing this intervention to play out over time with the 'correct' model is interesting and is almost certainly happening all of the time when new outbreaks occur. This idea is captured in the paper by the 'moving target' idea, which shows how large the error can get over time. One issue with this argument that is not properly accounted for in the paper is that these days studies will typically include an idea of the uncertainty of a forecast and the components of the model that are contributing to this uncertainty (e.g. demographic stochasticity, parameter uncertainty or, indeed, model uncertainty). When these are included, it's not clear that there would be such an easily measurable difference between the forecasts of two different models because these differences may be a) swamped by forecast uncertainties or b) ensembles of models may be used to make forecasts in any case. Some discussion of this should be included in the paper.

Another possible limit of the model is that it applies primarily to endemic infections, or epidemic infections with large attack rate, since the spatiotemporal dynamics of smaller outbreaks (and e.g. West African Ebola might be considered such) are often governed by stochastic effects rather than underlying inequality in susceptibility or contact patterns.

Reviewers' comments:

Reviewer #1 (Remarks to the Author):

The authors describe heterogeneity in TB risk and demonstrate that failure to introduce this heterogeneity in the analysis may lead to models missing their predictions in terms of relative impact of interventions. They introduce a metric to approximate the distribution of risk within a population (ie risk inequality coefficient), to allow modellers to use available (routinely collected) data to inform the needed distributions in risk and as an indicator for policy makers to assess progress in equalisation strategies. They compare their analysis in three countries with different patterns in TB epidemiology.

Response: Perfect!

Broadly, I find the article interesting for a general audience, especially those specialised in infectious disease control; however, in my opinion the authors tried to convey too many messages in the paper (methods, data availability, public health application, comparison across three countries) and, in doing so, it made their message muddled. I think the manuscript would benefit from refining the scope and focusing the message – is this a methodological or a public health message?

Response: The manuscript was originally written for Nature and hence had to comply with hard limits in length. The current revision has been considerably expanded as allowed by Nature Communications. We hope that this in itself has contributed to improve clarity and that the final paper will be enjoyed by many readers with diverse interests. Together with further modifications which also add to improving clarity (specified below), we strive to maintain the dual focus on methodology and public health, with a large overlap between the two. While we agree with the Reviewer that this is difficult, we would like to take on the challenge to produce a paper that can accomplish broad communication, although pending slightly more towards methodology but presented in a way that will be accessible to public health readers who might become essential facilitators of follow-up studies. Thanks to the advice already received from our Reviewers and Editor we believe this to be a realistic aim.

I find that the methods are appropriate and the authors have provided sufficient detail for other researchers to reproduce their work - if access is given to same NTP data. My only comment on the methods is that the authors compare future predictions using models with and without heterogeneity and find that these predictions differ. I am missing the uncertainty - are these predictions really different or is a homogeneous model a good-enough simplification of a heterogeneous one? A better way to test their hypothesis as to which model predicts better reality would be to compare with past trends.

Response: Uncertainly estimates are now provided. Besides being more informative this has also simplified the writing and enabled greater clarity by removing the need to present alternative scenarios. This is a major methodological addition to the paper. In response to the Reviewer's concern of whether the predictions made by heterogeneous and homogeneous models are significantly different, we have added tables confirming that credible intervals for parameters estimated by Bayesian MCMC are non-overlapping in all cases. Past incidence trends now have a much more crucial role in parameter estimation.

In terms of the conceptual development in methods - the authors made a compelling point to

advance way models represent populations when looking at TB transmission. As they point out, the concept of heterogeneity in risk affecting transmission and therefore the impact of interventions is not new. The novelty comes from trying to characterise this heterogeneity using routine data into a single metric. Would be interesting if the authors could place their approach in context - how does this method compare to efforts in other infectious diseases to characterise heterogeneity across geographies in transmission disease models, for example malaria and HIV? What are the advantages and intrinsic assumptions of using a GINI-like coefficient based on geographical distribution of risk?

Response: We are not aware of similar approaches in infectious diseases. This resembles, however, two recent studies in cancer which are cited in the manuscript (Mauguen & Begg 2016; Stensrud & Morten 2017). In infectious diseases, such as malaria and HIV, we found recent papers using Gini-like metrics:

Duarte EC, Ramalho WM, Tauil PL, Fontes CJF, Pang L. The changing distribution of malaria in the Brazilian Amazon, 2003-2004 and 2008-2009. *Rev Soc Bras Med Trop* 47, 763-769 (2014).

Christopoulos KA, Hartogensis W, Glidden DV, Pilcher CD, Gandhi M, Geng EH. The Lorenz curve: A novel method for understanding viral load distribution at the population level. *AIDS* 31, 309-310 (2017).

However, contrary to the cancer references these studies have different purposes from ours. The malaria paper is a descriptive study on the distribution of malaria in a country and makes no inferences about variation in individual risk. The HIV paper is concerned with the distribution of viral load (or infectiousness), precisely one of the characteristics that can be averaged without affecting model outputs. Since our manuscript is already extensive, we chose not to mention these studies to avoid more digression.

An addition we provide, however, is a comparison with metapopulation models to support a discussion on how our approach contrasts established practices in the modelling arena (expanded below).

The application of heterogeneity in risk concepts in models should be included for two main reasons: 1) failure to do so will lead to the wrong predictions of impact, and 2) it allows the monitoring of an indicator of inequality in risk distribution to track progress in equalising risk. The main assumption is that the aim of public health interventions is to equalise or at least change these distributions. However, not all risk inequalities are similar. Risk inequalities can arise for many reasons and can be considered fair (such as due to biological determinants) or unfair (such as due to poverty – this is mentioned in the introduction but it feels like a late addition and not an important point) from a public health perspective. Policy makers will want to monitor progress in equalising unfair distributions but may not want to allocate limited resources to other strategies focusing on equalising fair heterogeneity. In the paper, authors map risk distribution across districts.

The implication is that geographical distribution of risk is unfair. But without knowing what is driving this heterogeneity in geographical distribution, its correlation with socio economic status or poverty for example. it is difficult to assess the public health value and application.

Response: As above, the manuscript is already extensive and a fair treatment of equity aspects would deserve much more attention than we can allocate here. We fully agree with the Reviewer that the topic is important and timely, but we think it should be the subject of an entirely new article. To avoid confusing our readers, we now deliberately leave it out.

Regarding our use of geographical data we feel that a clarification need to be made. Our models account for two risk groups: a small proportion with high risk; and a vast majority with low risk. Since risk pertains to a probability of a response given a stimulus (i.e. acquiring disease given exposure to an infectious contact) it cannot be measured directly and must instead be inferred indirectly from patterns of incidence. In cases where individuals typically experience frequent disease episodes (such as influenza or malaria), direct measures of relative risk can be obtained from the frequencies at which individuals experience disease. But these individual-level measurements are not possible in TB due to the low frequency of the disease (it would require individuals living for millennia). For this technical reason we rely on grouping individuals by risk intensity. In this paper we adopt geographical groupings, for convenience given the available data, although it is not our intention to bind the study to geographical distributions as such (we could have chosen to group individuals by income, or some other variable likely to result in stratification of disease incidence). The way we conceive risk heterogeneity across districts is as if some districts have a higher frequency of high-risk individuals than others and for this reason incidence rates differ between districts. So, in essence, we conceive a risk distribution within each district, but our approach is agnostic of which individuals are high risk. So we are still some way from partitioning risk inequality into fair and unfair components as posed by the Reviewer, although this would be very worthwhile pursuing, we agree.

This rationale is now explicit in the form of a metapopulation model that we contextualise our model with. Hopefully this will not make the paper too heavy and the readers will find it useful and inspiring.

In terms of the public health message – I am not clear what was the advantage of applying this method to Vietnam, Brazil and Portugal. Was there an added value of cross country comparison? If so, it will be good to discuss that. I think the example of achieving TB targets in 2035 makes this work useful to policy makers; however, the assumptions on how to achieve the reductions needed in reactivation rates (among others) are unclear. The authors argue that it does not make a difference how you model these reductions – the point is that the predictions will change if heterogeneity is added to the model. But in doing so, they also reduce the public health value of the analysis.

Response: These are very crucial points. By including Vietnam, Brazil and Portugal we cover reasonably wide ranges of both TB burden and risk inequality levels. Regarding the assumptions on how to meet the targets, this has been hugely streamlined by the new statistical fittings. The model has essentially 3 parameters that can, in principle, be manipulated to reduce TB incidence: rate of successful cure; probability of progression from recent infection to active disease; and rate of reactivation of latent infection. Incidence trends alone do not allow the contributions of all three parameters to be estimated due to identifiability issues. In the original submission we dealt with this problem by making assumptions. In this revision, we incorporate more data which effectively resolves the identifiability issue. We use rates of successful treatment obtained from WHO reports to constrain the evolution of the treatment parameter and are left with only two parameters to estimate. The shape of past incidence trends enables the separation of reductions involving recent (primary) vs older (latent) infections with great certainty. We are very pleased with this result as we believe it greatly improves the public health value of our paper. It also leads to some interesting comparisons between the study countries. Brazil shows much greater reductions in reactivation rates, which seem to coincide with substantial investments in social protection. The greater prospects for TB control in Brazil identified in this study maybe inspiring for other countries.

Overall – I think it is a progressive manuscript that will advance the knowledge in the field, but the message needs more clarity and purpose and the methods need to include uncertainty estimates.

Response: Thank you, we hope to have addressed these concerns to the Reviewer's satisfaction.

Reviewer #3 (Remarks to the Author):

This work presents some interesting insights into the control of tuberculosis, and in the limitations of homogenous models in informing control priorities. However, I felt it was let down by being so hard to follow: I strongly suggest that the authors work to improve the clarity, in order to open this work to a public health audience, rather than limiting it to a specialist mathematical one.

Response: Sorry, the manuscript was originally written for Nature and hence had to comply with hard limits in length. The current revision has been considerably expanded as allowed by Nature Communications. We hope that this, together with other modifications implemented in response to the reviewers, has improved clarity and that the final paper will be enjoyed by many readers with diverse interests.

If I understand correctly, the main messages of the paper are:

1. Using homogeneous transmission models for heterogenous populations can distort the control priorities indicated by these models.

Response: Correct.

2. The extent of these distortions (and thus the utility of homogenous models) depends on the types of heterogeneity that are in effect.

Response: Yes.

3. In cases where there is large heterogeneity in acquisition of infection, homogenous models can overstate the importance of certain interventions, while understating the importance of others. However, other types of heterogeneity (e.g. in infectiousness) may be more benign.

Response: Yes, although we are now able to propose with simpler arguments that homogeneous models generally overstate the importance of interventions, but the opposite may occur in exceptional circumstances.

4. The authors present a method (analogous to Gini's coefficient in economics) for quantifying the amount of heterogeneity in a population structured in some way (e.g. geographically, or by some risk factor).

Response: Yes.

Main comments

The way the paper is currently structured, it appears that homogeneous models are inappropriate for any type of heterogeneity. However, it is only towards the end (p.8) that it becomes clear that this only applies under a very specific type of heterogeneity (in acquisition of infection). Under other types of heterogeneity, presented only in the supporting information, there is less distortion between homogenous and heterogeneous models. The authors should make this much clearer - perhaps bringing some of those supporting information figures to the main text, to enable a fair comparison of different types of heterogeneity. Wording also needs to be adjusted in places: for example in p.2 line 37, where 'unsuitable' should be 'potentially unsuitable'. Acknowledging this also highlights a key data need, understanding what type of heterogeneity is operating in a given population.

Response: All model variants are now in the main text. We fully agree that this makes a much better paper.

I found the references to 'cohort selection' much more confusing than clarifying: it needs to be either rewritten, moved to the supporting information, or removed altogether. My issue is that for TB, reinfection can and does occur. That is, why should it matter whether or not the disease preferentially attacks the 'high risk' group? They can be reinfected anyway. My suggestion is that this might be better explained in terms of the effective reproduction number: adverse changes in population susceptibility should, in principle, manifest through this metric. At the moment though, to me figure 3 does very little to clarify the dynamics.

Response: This has been modified in depth. The term "cohort selection" is no longer used but we maintain that selection acts on individual risk whenever a force of infection is in operation. The occurrence of reinfection does not disable this process. In diseases where reinfection occurs, higher risk individuals spend more time being diseased or under treatment and therefore removed from susceptible groups. The occurrence of selection is not only postulated, but actually monitored as the model runs (plots and numbers representing mean risks in specific epidemiological compartments are provided as part of Figs. 2, 3 and 7). The manuscript has been entirely rewritten and we believe the new version to be much more reader friendly.

The use of the Lorentz curve is an innovative and revealing approach to inequalities in TB. However, the authors should also acknowledge its limitations more fully: especially, that it is best suited for situations where almost all cases are notified - otherwise it would tend to obscure, not reveal, some important inequalities (principally in access to high-quality, TB-notifying services).

Response: This is now addressed in the Discussion.

Other comments

Although the Gini coefficient is a fresh and interesting proposal, straightforward statistical tests also detect population variation in risk (e.g. routine generalized linear models). It will be helpful to clarify what, if at all, the Lorentz curve approach offers in addition to these straightforward methods. Otherwise, it seems to me to be a useful and helpful way of communicating (but not necessarily detecting) inequality in a population. There is also a need

to state more clearly the limitations of the 'Lorentz approach', as described below.

Response: The vast use of Lorenz curves and Gini metrics in economics makes their introduction in health research and policy appealing. Not only communication across the disciplines will be facilitated by the adoption of common concepts, but also specific comparisons (either directly or through integrative methodological development) will become possible.

Authors have calibrated both models (heterogeneous and homogeneous) to give the same incidence. Do the models also agree in prevalence? This would seem important as the prevalence/incidence combination determines the average duration of disease, which in turn can be critical for intervention priorities. Have the authors controlled for this variation by ensuring both models are matched by prevalence too?

Response: When the model is at equilibrium (which we had to assume for initiation in 2002) agreement in incidence implies agreement in prevalence as well. As incidence declines from 2002 onwards we have not monitored agreement in prevalence between models (as we did not have availability of prevalence data) but see no reason to expect noticeable deviations given that they started by being coincident in 2002. In relation to this issue, however, we find heterogeneous models to agree better than homogeneous with prevalence of latent TB infection as estimated by an independent study (Houben & Dodd PLOS Med. 13, e1002152 (2016)).

In practice, contact heterogeneity can play an important role in interactions between 'high-risk' and 'low-risk' individuals - potentially weakening the effects demonstrated in this paper, in cases where individuals preferentially mix with others in their own risk group. It would be helpful to describe this as a limitation.

Response: We have not investigated this but agree that it will affect quantitatively (but not qualitatively) the effects we describe. This is now specified in Methods.

The manuscript would benefit greatly from editing to improve clarity and to help the reader. Some examples:

- Comments are made without explanation, e.g. reference to 'variance' without defining what variance is being calculated, and the 'conservative' nature of these estimates, without explaining why or in what sense they're conservative

Response: 'Variance' is now expressed as $\text{var}(\alpha)$ and the term 'conservative' is no longer used.

- p.4, line 81: Suggest avoid mentioning the model here - this paragraph refers to the inequality metric that the authors propose, which later serves as an input for the modelling. Mentioning the model at this stage seems to confuse things.

Response: This was required in the original manuscript because the then Fig. 1 was a composite of model and inequality metric. Such composite figure is no longer included which naturally simplified the text in question.

- p.4, line 89: "...as the single most important process..." Suggest to reword this so it's clear that these are modelled scenarios we're talking about, rather than measured phenomena.

Response: This has been rewritten as: "Model outputs were analyzed in-depth revealing a poor predictive capacity of homogeneous models and leading to the identification of acquisition of infection as the single most important process behind model disparities."

- p.5, line 109: Here and in the Methods, it is unclear what it means to 'share equally' incidence declines amongst different parameters.

Response: This is no longer done (nor written) because we were able to estimate the specific contributions of each parameter to the incidence decline.

- p.6, line 123: "...adjust the data undistinguishably well..." Suspect there's a typo here - did the authors intend to say 'capture the data equally well' or something to that effect?

Response: We have adopted the phrase and thank the Reviewer.

- p.11, equation 5: until this point I had understood that the model was about heterogeneity in susceptibility to infection, but it now appears for the first time that it additionally includes heterogeneity in transmission. Again, this confuses things: to make the story more systematic (and easier to follow), the authors could, for example, present a range of different types of heterogeneity in the main text, showing a comparison between them. Or, in the main text, they might show the model with solely heterogeneity in susceptibility, and demonstrate the other mechanisms in the supporting info. The current approach, which mixes these approaches, is confusing.

Response: As above we now present all model variants in the main text. We see this as a major improvement and, again, thank the Reviewer from bring it up.

- p.12, line 259: I'm not sure why annual incidence is being calculated from the point values of P_i , in a non-equilibrium model - isn't there an integral term needed here?

Response: We approximate incidence as the sum of the positive terms in the rate of change dI/dt . Given that the time units in the model are years we find this approximation to be adequate.

- p.13, line 292: Crucial, but obscure - please clarify, or remove.

Response: This is now extensively explained in the Results section.

Figures and tables

Again, I found these outputs unnecessarily hard to follow.

Figure 1c. What do the histograms show? The caption suggests that the figure illustrates the distribution in parameters, but the histogram appears to be associated with the state variables. It is also unclear what is meant by 'segregated by the transmission dynamics calibrated by the 2002 incidence'.

Response: As above, this is now extensively described.

Figure 2. Please add a legend to highlight at a glance which parameters the different colours represent, even if only in panel c, or any other panel with available whitespace. Moreover, why does the blue curve show such abrupt changes in incidence (decreasing and then levelling off) compared to the other curves? Also, please mention briefly in the figure caption, how the ‘scale changes’ were selected for each of the parameters after 2020.

Response: This figure is now much simpler as we were able to estimate parameters and no longer need to present alternative scenarios.

Figure 3. Is this the same as the ARTI, or equivalently the force-of-infection? If so, why should latent and uninfected compartments suffer different forces-of-infection? I suppose this may have something to do with the fact that there tends to be an over-representation of high-risk individuals in the latent compartment - in which case it would be really helpful for the reader to mention this in the caption. Also, to aid comparison, it would help to label each country with a different colour, perhaps distinguishing the latent and uninfected by solid and dashed lines. I’m not sure what the current dashed lines add to the figure.

Response: Sorry for the confusion generated by this figure. It has been essentially removed. The only reminiscent of the selection processes shown there are the secondary panels in the new figures 3(a,c,e) and 7(b) – each displaying only a single curve.

Extended Data Fig.2: What does this add to Figs 1 and 4? Could the ratio of A and B terms not be explained easily in Fig.1 instead?

Response: No longer shown.

Extended Data Fig.3: Please clarify by adjusting the caption to ‘Risk inequality metrics under alternative administrative groupings’ or similar.

Response: We adopted the phrase and thank the Reviewer.

Extended Data Fig.5: seems important to the story, but rather mysterious in its current form. What are the parallel diagonal lines in panel A? What are the circles? These don’t seem to be described anywhere that is straightforward to find.

Response: This has been moved to the main text with more discussion and replicated for the other countries. The procedure is detailed in Methods and Supplementary Table 1.

Extended Data Fig.6: What is a ‘cumulative reduction’ here - may be better to call this the ‘reduction relative to 2020 value’ or something similar?

Response: This figure is no longer shown.

- p.31, Extended data table 1: Please clarify what the semi-colons mean

Response: Done.

- p.32, Extended data table 1: the reader is left to track down what r_w means - it would be

helpful to add a brief description here.

Response: Done.

Reviewer #4 (Remarks to the Author):

A very interesting paper, with simple yet powerful ideas written up in a clear way. The idea of a Gini coefficient, of the kind used routinely by economists, for infectious disease epidemiology is relevant and could potentially provide a very compact summary of the degree of heterogeneity in transmission seen in populations. The authors provide model based evidence to demonstrate how this measure could lead to significantly different calculations of quantities that are of enormous importance such as the estimated time required for success of an intervention program.

Response: Thank you!

I do think that it's currently harder than it needs to be in the paper to see how to apply the Gini coefficient, and implied Lorenz curve, to heterogeneous risk group mathematical models in order to allow projections of the impact of interventions to be made. Figure 1 is good and almost contains enough information for a modeler to reconstruct the calculations of the authors, but it would be good if the authors could include an example calculation (perhaps in supplementary material) in which the parameters α_1 and α_2 are discovered via the process described in the methods section.

Response: This figure has been decomposed into what are now figures 1 and 2 and considerably more detail and discussion is provided in associated with its construction. Hopefully this contains the explanation that the Reviewer is looking for. α_1 and α_2 are essentially the result of an exact calculation which we approach by simple numerical search.

Another issue that deserves comment by the authors (if not a calculation) is whether a two subgroup model is a sufficiently finely-structured model to be used as a simple proxy for a heterogeneous model in which the full heterogeneity implied by the Lorenz curve is modeled (perhaps by a large number of population subgroups that would more accurately reflect the true susceptibility distribution). I'm not suggesting that such a model needs to be constructed by the authors here, but that there could be some explanation of the sufficiency of the two group model.

Response: We have slightly changed the approach in a way that hopefully resolves this concern. We now begin by replacing the original Lorenz curves with coarser 2-segment curves with the same RIC (figure 1) and use these directly to inform the 2-risk-group model. More finely resolved distributions are shown in Supplementary Figure 5 but not used for modeling. We also contextualize our 2-group models with slightly more resolved metapopulations in the discussion to further explore the mechanisms enabled by risk distributions. This is only described very briefly as the paper is already dense, but we believe it points to a whole new direction for future research.

The idea of estimating the necessary intervention strength using an incorrect model and then allowing this intervention to play out over time with the 'correct' model is interesting and is almost certainly happening all of the time when new outbreaks occur. This idea is captured in

the paper by the 'moving target' idea, which shows how large the error can get over time. One issue with this argument that is not properly accounted for in the paper is that these days studies will typically include an idea of the uncertainty of a forecast and the components of the model that are contributing to this uncertainty (e.g. demographic stochasticity, parameter uncertainty or, indeed, model uncertainty). When these are included, it's not clear that there would be such an easily measurable difference between the forecasts of two different models because these differences may be a) swamped by forecast uncertainties or b) ensembles of models may be used to make forecasts in any case. Some discussion of this should be included in the paper.

Response: This figure (now figure 3) was included as an illustration of the mechanism by which homogeneous models are expected to deviate from realistic (heterogeneous) population dynamics. It was produced such that all points could be calculated exactly so attention would not be distracted by other issues, which are certainly important but different. Real data starts being implied from figure 4 onwards, and there we adopt a Bayesian MCMC scheme to estimate model parameters in this new version of the manuscript. The procedures represented in the two figures are not exactly the same: in figure 3 we have one parameter to fit one point so the calculation is exact (ie no parameter uncertainty); while in figure 4 we have two parameters to fit a series of 13 points, but the credible intervals are so narrow that the role of parameter uncertainty appears negligible. This is now described in context with the respective figures in the manuscript. Stochasticity is deliberately not considered in this study but it opens important points when contrasted with heterogeneity (model uncertainty) which are now integrated in the Discussion.

Another possible limit of the model is that it applies primarily to endemic infections, or epidemic infections with large attack rate, since the spatiotemporal dynamics of smaller outbreaks (and e.g. West African Ebola might be considered such) are often governed by stochastic effects rather than underlying inequality in susceptibility or contact patterns.

Response: We take this as an interesting direction for further research!

Reviewers' comments:

Reviewer #1 (Remarks to the Author):

I would like to thank the authors for the extensive revision of this paper. their additions and edits have made the manuscript clearer and, in my opinion, more useful. I feel they have successfully addressed my comments.

Reviewer #3 (Remarks to the Author):

I appreciate the changes that the authors have made, and still feel that the core ideas in this paper will be of interest to a broad audience. Unfortunately, however, there remain real issues in the presentation, making it unnecessarily hard work for a reader to appreciate these ideas. Given the likely interest in this work, I would strongly encourage the authors to consider accommodating real changes to the presentation (not just minor adjustments), to address these concerns for the benefit of a broad audience.

Main comments

>> The use of Bayesian MCMC for parameter estimation was puzzling at first glance, but I recognize that the authors included this in response to a reviewer comment. That comment made the valid point that variation between model forecasts may be small in relation to the uncertainty in respective model projections. The authors have attempted to address this comment by including a Bayesian MCMC, finding that uncertainty in model projections are trivially small. However, neither this finding, nor the approach, are consistent with typical modelling approaches to uncertainty. For example, the current study does not include uncertainty in parameter values, a pervasive feature in other Bayesian modelling analyses (for example, Menzies et al, PLoS Med 2012, and Marx et al, Lancet GH 2018). Moreover, while the present study includes a 'measurement error' for the notifications, no details are given for the assumed size of this error (sigma in equation 7), nor for the dispersion on the Gaussian priors, nor for the extent to which the resulting MCMC is successfully 'mixed'.

Overall therefore, the Bayesian inference serves only to confuse the narrative, without adequately addressing the initial point about uncertainty. I suggest that the authors either (i) consider implementing a 'proper' Bayesian MCMC, i.e. incorporating parameter uncertainty; incorporating 'true' uncertainty in the proportion of cases that are notified (WHO estimates certainly have appreciable ranges in these estimates); as well as including model outputs to help assess the quality of the traces, or (ii) exclude uncertainty altogether, instead leaving it to the discussion to comment that in practice, differences between hetero- and homogeneous models may be subsumed by model uncertainty. A third way might be to estimate the maximum model uncertainty under which the models would still be distinguishable, and to compare this threshold against model projections in previous studies.

>> p.8, line 161: 'The rate of successful treatment...' The WHO data shown in Suppl. Figure 2 represents the proportion of treatment initiations that successfully completed treatment. While this may not have shown any trends in recent years, it is entirely possible that declining trends have arisen because of gradual improvements in routine TB diagnosis and access to TB services, i.e. decreasing beta and increasing tau, without any secular change in the treatment success rates. In fact, changes in beta are the approach most often used in other models (for example, see Salje et al, PLoS Med 2014). Please replace this sentence by acknowledging that these are another possible mechanism for the declines in incidence (and indeed for future projections).

>> Related to the above point, there is a tendency for the model narrative to lean heavily on the

mechanisms being modelled in this paper, to suggest that the essential modelling results (that homogenous models tend to underestimate the intensity of control efforts) will apply more generally in TB transmission models. However, as noted above, the current study is simply too narrow to make this claim. Thus, for example, I suggest that the authors exercise more caution in extended discussions such as on p.9, which – as well as distracting from the interesting primary results – also give the (flawed) impression that these results arise from some deep dynamical behaviour that is generalisable across models. It would be scientifically robust to replace these discussions altogether, with a simple acknowledgement that other mechanisms may give qualitatively different results.

Other comments

The following are additional areas where I struggled, and strongly appeal to the authors to make the manuscript (and its nice ideas) much more accessible.

p.4, lines 64 – 75: Although worded as a new concept, we know that there have been many observational studies identifying TB risk factors (e.g. for diabetes, smoking, malnutrition, etc etc). This part of the narrative should therefore be clarified to indicate how (if at all) this concept is different, or dropped altogether.

p.4, line 95: The text suggests that the reason for adopting a 4%/96% categorisation is for consistency with previous studies – is that correct? If so, surely it's more important to be consistent with the empirical Lorentz curves that the authors display in Fig.1 (at present the solid and dashed lines look very different). Otherwise, it is not clear how those curves, a key part of the study, actually inform the modelling. Ideally, the dashed lines should be chosen at least to replicate the empirical Gini coefficients shown in Fig.1 – and this should be explained clearly in the Methods, so that the reader can understand that the empirical data is indeed informing the modelling.

p.5, line 99: '...there is no reason to expect...' Not clear that this is true, and indeed this distracts from the main point of the study, which is to illustrate model performance under a given type of heterogeneity (see also above point on generalisability). Suggest to drop this clause, which raises more issues than it addresses.

p.6, line 113: 'As our procedures involve models...' I am afraid this sentence makes no sense to me, perhaps because it is condensing a lot of information into a few words. It sounds like an explanation of the calibration process, in which case perhaps it's better explained at greater length (and clarity) in the supporting information.

p.6, line 120: Please clarify this data. Presumably it is for reported TB cases (notifications)? What is its source? Is it from 2002, which is when the calibrated model simulations are initiated from?

p.6, line 122: 'Notice that these variances are consistently higher...' This is unclear. First, I assume the 'observed variances in TB incidence' refer to the variation across admin divisions shown in Fig.1? (Please clarify) If so, why should we expect these to be comparable with the alpha-variances, given that there is a nonlinear transmission model mediating the two?

p.6, line 133: This explanation is still opaque to me. It is difficult to see why the uninfected compartment is presented as being so important for transmission, given that the other compartments are able to be infected as well. I suggest this interpretation does not add to the paper, only detracts from it. Moreover, this whole narrative seems to be geared towards arguing for the generality of the results in this paper. However, see the main comment above on generalisability: it would be scientifically more robust to simply acknowledge that other mechanisms may be equally valid, potentially with different conclusions. In which case I suggest

there is no need for the confusing narrative here on 'selection'.

p.9, line 192: '...have been predominantly attributed...' See comment above, on the quality of the parameter fits. Without more information, it is hard to know the extent to which these are unique parameter fits – there may be other values of r_{ϕ} , r_w that match the data equally well, and that place different weights on these parameters (given that the calibration is matching incidence trends alone, it would not be surprising if such alternative parameter fits indeed exist).

p.14, line 313: This is not the usual interpretation of stochasticity (at least in epidemiology), and it need not pertain to an individual – an example is household transmission, which can be stochastic. Moreover, it seems odd to claim that stochasticity and heterogeneity are often confused (at least in epidemiology – the cited source refers to an ecological paper): I don't expect there are modellers who would deny the existence of stochastic, homogenous populations. Please rephrase to clarify.

Figure 2 seems more complex than needed – for example, what are the numbers in square brackets? It seems the information here would be much clearer in a table, either replacing or supporting the figure.

Figure 4. Please explain what the numbers next to multiplication signs are (i.e. the values of κ , where κ is only mentioned in the main text).

Reviewer #4 (Remarks to the Author):

The authors have done a good job of responding to reviewers' comments.

Reviewers' comments:

Reviewer #1 (Remarks to the Author):

I would like to thank the authors for the extensive revision of this paper. their additions and edits have made the manuscript clearer and, in my opinion, more useful. I feel they have successfully addressed my comments.

Response: Thank you!

Reviewer #3 (Remarks to the Author):

I appreciate the changes that the authors have made, and still feel that the core ideas in this paper will be of interest to a broad audience. Unfortunately, however, there remain real issues in the presentation, making it unnecessarily hard work for a reader to appreciate these ideas. Given the likely interest in this work, I would strongly encourage the authors to consider accommodating real changes to the presentation (not just minor adjustments), to address these concerns for the benefit of a broad audience.

Main comments

>> The use of Bayesian MCMC for parameter estimation was puzzling at first glance, but I recognize that the authors included this in response to a reviewer comment. That comment made the valid point that variation between model forecasts may be small in relation to the uncertainty in respective model projections. The authors have attempted to address this comment by including a Bayesian MCMC, finding that uncertainty in model projections are trivially small. However, neither this finding, nor the approach, are consistent with typical modelling approaches to uncertainty. For example, the current study does not include uncertainty in parameter values, a pervasive feature in other Bayesian modelling analyses (for example, Menzies et al, PLoS Med 2012, and Marx et al, Lancet GH 2018). Moreover, while the present study includes a ‘measurement error’ for the notifications, no details are given for the assumed size of this error (sigma in equation 7), nor for the dispersion on the Gaussian priors, nor for the extent to which the resulting MCMC is successfully ‘mixed’.

Overall therefore, the Bayesian inference serves only to confuse the narrative, without adequately addressing the initial point about uncertainty. I suggest that the authors either (i) consider implementing a ‘proper’ Bayesian MCMC, i.e. incorporating parameter uncertainty; incorporating ‘true’ uncertainty in the proportion of cases that are notified (WHO estimates certainly have appreciable ranges in these estimates); as well as including model outputs to help assess the quality of the traces, or (ii) exclude uncertainty altogether, instead leaving it to the discussion to comment that in practice, differences between hetero- and

homogeneous models may be subsumed by model uncertainty. A third way might be to estimate the maximum model uncertainty under which the models would still be distinguishable, and to compare this threshold against model projections in previous studies.

Response: In line with what the Reviewer implies above we also believe that a full treatment of uncertainty is beyond the scope of the current paper, which is mainly about introducing a new approach to the design of infection transmission models that capture heterogeneity in risk among individuals in a population. Incidence data from 3 countries are utilized to inform heterogeneous models as well as their mean-field approximations. Models are analysed side-by-side to describe the essence of the divergence between the two formulations in a rather exhaustive fashion and the procedure is contrasted with common practices in the field. Altogether this amounts to around 20 figures and 10 tables (including those in supplementary information) alongside the introduction of concepts from other disciplines which are, in our view, not easy to assimilate. With a full treatment of uncertainty we would risk to lose the attention of our readers away from the key messages.

However, addressing uncertainty was not the only reason for adopting the MCMC approach. In the first round of reviews, Reviewer 1 criticised the then arbitrary assumption of splitting the incidence decline equally among three parameters (ϕ , ω , τ). To address this criticism we decided to implement a parameter estimation procedure and let the procedure tell us how the decline was shared among specified sets of parameters. We chose MCMC tools readily available in MATLAB to do this as now more detailed in the Methods section. It has never been our intention to estimate all the parameters in the model and respective uncertainties, but only the rates of decline in specified parameters. To avoid making the paper inference-heavy we restricted our estimations to sets of 2 parameters only. We think that this has improved the paper and would like to leave it.

As a compromise we have replaced the discussion around the confidence intervals obtained with this particular procedure by a more general discussion of uncertainty from our perspective. We have also specified our assumptions concerning measurement error and provided outputs for assessing the quality of the traces in one of the scenarios presented (Methods and Supplementary Figures 9 and 10). Other scenarios perform similarly.

>> p.8, line 161: ‘The rate of successful treatment...’ The WHO data shown in Suppl. Figure 2 represents the proportion of treatment initiations that successfully completed treatment. While this may not have shown any trends in recent years, it is entirely possible that declining trends have arisen because of gradual improvements in routine TB diagnosis and access to TB services, i.e. decreasing β and increasing τ , without any secular change in the treatment success rates. In fact, changes in β are the approach most often used in other models (for example, see Salje et al, PLoS Med 2014). Please replace this sentence by acknowledging that these are another possible mechanism for the declines in incidence (and indeed for future projections).

Response: The Reviewer is making some important points here. We are no longer using the data on rates of successful treatment to inform τ . Instead we allow τ to vary as well as β as the Reviewer suggests. As explained above we consider sets of no more than 2 parameters to vary per scenario but we run several alternative scenarios: (i) ω only; (ii) ϕ and ω ; (iii) τ and ω ; (iv) β and ω . This was revealing and enabled us

to make the paper more comprehensive.

>> Related to the above point, there is a tendency for the model narrative to lean heavily on the mechanisms being modelled in this paper, to suggest that the essential modelling results (that homogenous models tend to underestimate the intensity of control efforts) will apply more generally in TB transmission models. However, as noted above, the current study is simply too narrow to make this claim. Thus, for example, I suggest that the authors exercise more caution in extended discussions such as on p.9, which – as well as distracting from the interesting primary results – also give the (flawed) impression that these results arise from some deep dynamical behaviour that is generalisable across models. It would be scientifically robust to replace these discussions altogether, with a simple acknowledgement that other mechanisms may give qualitatively different results.

Response: Selection within cohorts is well described in demographic research and produces general effects in infection transmission models. To support this statement we refer the Reviewer to a short invited piece which was part of a celebration of 2018 as the year of biomathematics. To address the comments more specifically, we have added 3 new scenarios to describe what might have been causing the observed incidence declines. We cannot overstate how grateful we are for having been led to make this extension. As the Reviewer suspected we were not capturing all the possible interpretations. Varying the combinations (τ, ω) or (β, ω) leads to consistent conclusions between the two but somewhat different from varying (ϕ, ω). This is now documented in the paper together with a discussion on the need for discriminatory data to enable testing the alternative hypotheses. The one aspect where we do not agree with the Reviewer, however, is in the statement that this does not come from deep dynamical behaviour as we believe to have provided intuitive mechanistic interpretations for all our findings – all rooted in the same selection process!

Other comments

The following are additional areas where I struggled, and strongly appeal to the authors to make the manuscript (and its nice ideas) much more accessible.

Response: This is very much appreciated. Below we detail some of the internal machinery of our models (much more than minimalistically). The Reviewer is obviously very knowledgeable of TB modelling and this interaction already stands as a great example of how in-depth peer review improves accessibility of a paper intended for a wide audience. Thank you!

p.4, lines 64 – 75: Although worded as a new concept, we know that there have been many observational studies identifying TB risk factors (e.g. for diabetes, smoking, malnutrition, etc etc). This part of the narrative should therefore be clarified to indicate how (if at all) this concept is different, or dropped altogether.

Response: We are dealing with unobserved heterogeneity rather than measured risk factors. The idea is that the risk of each individual is determined by many unobserved factors which differ among individuals creating some level heterogeneity that cannot be fully measured directly. However, this heterogeneity affects patterns of disease incidence in populations and we propose to work in reverse, i.e. we propose to use incidence patterns to infer the

underlying unobserved heterogeneity. This has been made more explicit in the paragraph just before the passage quoted by the Reviewer: “The premise is that variation in the risk of acquiring a disease (whether infectious or not) goes beyond what is captured by measured factors (typically age, malnutrition, comorbidities, smoking habits, social contacts, etc), and a distribution of unobserved heterogeneity can be inferred from incidence trends in a holistic manner.” We present one scheme for this estimation but hope that this paper will stimulate other researchers to think along these lines and formulate more and better schemes.

p.4, line 95: The text suggests that the reason for adopting a 4%/96% categorisation is for consistency with previous studies – is that correct? If so, surely it’s more important to be consistent with the empirical Lorentz curves that the authors display in Fig.1 (at present the solid and dashed lines look very different). Otherwise, it is not clear how those curves, a key part of the study, actually inform the modelling. Ideally, the dashed lines should be chosen at least to replicate the empirical Gini coefficients shown in Fig.1 – and this should be explained clearly in the Methods, so that the reader can understand that the empirical data is indeed informing the modelling.

Response: The dashed lines are the 96:4 partition that has the same Gini coefficient as the solid lines. We need a 2-segment approximation of the solid curve in order to use a 2-group model. Each solid curve is made of hundreds or thousands of tiny segments (coinciding with the number of administrative divisions in the data), and using such resolution would require a model with the same number of risk groups as segments in the curve. This can be done but it would result be a very high dimensional model which for a first introduction of the approach could distract from the main point. Instead we adopted a 2-group model with the same risk-variance (same Gini coefficient) as the original problem. We edited the Methods sections for improved clarity around this point.

p.5, line 99: ‘...there is no reason to expect...’ Not clear that this is true, and indeed this distracts from the main point of the study, which is to illustrate model performance under a given type of heterogeneity (see also above point on generalisability). Suggest to drop this clause, which raises more issues than it addresses.

Response: Thank you, we have removed this clause.

p.6, line 113: ‘As our procedures involve models...’ I am afraid this sentence makes no sense to me, perhaps because it is condensing a lot of information into a few words. It sounds like an explanation of the calibration process, in which case perhaps it’s better explained at greater length (and clarity) in the supporting information.

Response: This text has been removed.

p.6, line 120: Please clarify this data. Presumably it is for reported TB cases (notifications)? What is its source? Is it from 2002, which is when the calibrated model simulations are initiated from?

Response: This query concerns the inference of risk variances from stratified incidence data. The paragraph above the line quoted by the Reviewer directs the reader to the Methods section where the procedure is explained (now modified for improved clarity). At this point we trust that interested readers have seen the description in Methods and refrain from repeating it in the main text. In relation to this, we have also simplified the description of the metapopulation models in the Discussion which should also help clarify how risk variances are conceived and treated.

p.6, line 122: ‘Notice that these variances are consistently higher...’ This is unclear. First, I assume the ‘observed variances in TB incidence’ refer to the variation across admin divisions shown in Fig.1? (Please clarify) If so, why should we expect these to be comparable with the alpha-variances, given that there is a nonlinear transmission model mediating the two?

Response: The observed variances in TB transmission refer to the variation across divisions (data which led to the construction of the Gini curves) or, equivalently, to its reorganization into 2 segments as clarified above. The reviewer is right in commenting that there is a nonlinear model mediating the alpha-variances and the incidence-variances, but here we emphasise a phenomenon that is essentially linear and makes variances appear smaller than they actually are. Consider, for example, a simple SIR model with 2 risk groups and constant force of infection (this model is linear):

$$\begin{aligned}\frac{dS_1}{dt} &= q_1\mu - \alpha_1\lambda S_1 - \mu S_1 \\ \frac{dI_1}{dt} &= \alpha_1\lambda S_1 - (\tau + \mu)I_1 \\ \frac{dS_2}{dt} &= q_2\mu - \alpha_2\lambda S_2 - \mu S_2 \\ \frac{dI_2}{dt} &= \alpha_2\lambda S_2 - (\tau + \mu)I_2,\end{aligned}$$

with $q_1 + q_2 = 1$, $q_1\alpha_1 + q_2\alpha_2 = 1$ and $var(\alpha) = q_1(\alpha_1 - 1)^2 + q_2(\alpha_2 - 1)^2$ as in the paper. This model has an endemic equilibrium given by

$$\begin{aligned}I_1 &= \frac{\alpha_1\lambda q_1\mu}{(\tau + \mu)(\alpha_1\lambda + \mu)} \\ I_2 &= \frac{\alpha_2\lambda q_2\mu}{(\tau + \mu)(\alpha_2\lambda + \mu)}.\end{aligned}$$

The group-specific incidences are then calculated and depicted in the top panel of the figure below as functions of the force of infection λ , for a particular $var(\alpha) = 4$ chosen for illustrative purposes. To facilitate comparison of the alpha-variances with the respective incidence-variances we normalise the group-specific incidence-variances to obtain a distribution with mean 1 as in the paper. The variances of the normalised incidences (Y_1 and Y_2) are then calculated and plotted in the bottom panel together with the alpha-variance. They are consistently lower because the susceptibility pool of the higher-risk group is depleted faster. This selection phenomenon is the heart of our paper.

This model reproduces the tendency for alpha-variances being consistently higher than the respective incidence-variances highlighted by the reviewer and it is only a linear model with selection on susceptibility. Naturally, if the model is made nonlinear by introducing an explicit dependence of λ on the prevalence of infection the result is maintained, but this is not conditional on nonlinearity.

p.6, line 133: This explanation is still opaque to me. It is difficult to see why the uninfected compartment is presented as being so important for transmission, given that the other compartments are able to be infected as well. I suggest this interpretation does not add to the paper, only detracts from it. Moreover, this whole narrative seems to be geared towards arguing for the generality of the results in this paper. However, see the main comment above on generalisability: it would be scientifically more robust to simply acknowledge that other mechanisms may be equally valid, potentially with different conclusions. In which case I suggest there is no need for the confusing narrative here on ‘selection’.

Response: An identical argument can be made about individuals being infected out of the latent compartment and this is acknowledged in the paper. Now more emphatically: “A similar process occurs for other epidemiological compartment where individuals are at risk of infection – uninfected (U) and latent (L) in the case of the model adopted here.” In Fig 3, for example, panels a, c, e show the effects of weakening selection (mediated by the force of infection) occurring in both U and L compartments. As stated above selection is the heart of our paper, so it cannot be removed. We could more easily replace TB by malaria or HIV, for example, than remove selection from the narrative.

p.9, line 192: ‘...have been predominantly attributed...’ See comment above, on the quality of the parameter fits. Without more information, it is hard to know the extent to which these are unique parameter fits – there may be other values of r_{ϕ} , r_w that match the data equally well, and that place different weights on these parameters (given that the calibration is matching incidence trends alone, it would not be surprising if such alternative parameter fits indeed exist).

Response: We have reduced the weight on these particular parameter estimates by including more estimated parameter scenarios and acknowledging that the results of these inferences should be treated as hypotheses subject to further testing (detailed above).

p.14, line 313: This is not the usual interpretation of stochasticity (at least in epidemiology), and it need not pertain to an individual – an example is household transmission, which can be stochastic. Moreover, it seems odd to claim that stochasticity and heterogeneity are often confused (at least in epidemiology – the cited source refers to an ecological paper): I don't expect there are modellers who would deny the existence of stochastic, homogenous populations. Please rephrase to clarify.

Response: This paragraph has been removed.

Figure 2 seems more complex than needed – for example, what are the numbers in square brackets? It seems the information here would be much clearer in a table, either replacing or supporting the figure.

Response: The numbers represent mean(α) in each epidemiological compartment. This is now explicit in the figure legend. We recognise that the intent of the figure could not be appreciated without clarity about what the numbers in bracket represent and hope that this clarification settles the issue.

Figure 4. Please explain what the numbers next to multiplication signs are (i.e. the values of κ , where κ is only mentioned in the main text).

Response: Good point! This is now explicit in the figure legends.

Reviewer #4 (Remarks to the Author):

The authors have done a good job of responding to reviewers' comments.

Response: Thank you!

REVIEWERS' COMMENTS:

Reviewer #3 (Remarks to the Author):

My thanks to the authors for accommodating the extra clarifications requested on the manuscript. Just as one final (and entirely optional) suggestion, readers who are not familiar with the role of 'selection' may find it helpful to have a citation pointing to the source that the authors attached in response to this latest round of comments, 'on the mathematics of populations', as further background.

REVIEWERS' COMMENTS:

Reviewer #3 (Remarks to the Author):

My thanks to the authors for accommodating the extra clarifications requested on the manuscript. Just as one final (and entirely optional) suggestion, readers who are not familiar with the role of 'selection' may find it helpful to have a citation pointing to the source that the authors attached in response to this latest round of comments, 'on the mathematics of populations', as further background.

Response: We appreciate the Reviewer's suggestion for including a citation to the biomathematics commemoration piece "On the Mathematics of Populations". This is now included in the reference list as a biorxiv preprint.